# Pleistocene sediment DNA reveals hominin and faunal turnovers at Denisova Cave

Elena I. Zavala[1✉], Zenobia Jacobs[2,3✉], Benjamin Vernot[1], Michael V. Shunkov[4], Maxim B. Kozlikin[4], Anatoly P. Derevianko[4], Elena Essel[1], Cesare de Fillipo[1], Sarah Nagel[1], Julia Richter[1], Frédéric Romagné[1], Anna Schmidt[1], Bo Li[2,3], Kieran O'Gorman[2], Viviane Slon[1,5,6,7], Janet Kelso[1], Svante Pääbo[1], Richard G. Roberts[2,3✉] & Matthias Meyer[1✉]

Denisova Cave in southern Siberia is the type locality of the Denisovans, an archaic hominin group who were related to Neanderthals[1–4]. The dozen hominin remains recovered from the deposits also include Neanderthals[5,6] and the child of a Neanderthal and a Denisovan[7], which suggests that Denisova Cave was a contact zone between these archaic hominins. However, uncertainties persist about the order in which these groups appeared at the site, the timing and environmental context of hominin occupation, and the association of particular hominin groups with archaeological assemblages[5,8–11]. Here we report the analysis of DNA from 728 sediment samples that were collected in a grid-like manner from layers dating to the Pleistocene epoch. We retrieved ancient faunal and hominin mitochondrial (mt) DNA from 685 and 175 samples, respectively. The earliest evidence for hominin mtDNA is of Denisovans, and is associated with early Middle Palaeolithic stone tools that were deposited approximately 250,000 to 170,000 years ago; Neanderthal mtDNA first appears towards the end of this period. We detect a turnover in the mtDNA of Denisovans that coincides with changes in the composition of faunal mtDNA, and evidence that Denisovans and Neanderthals occupied the site repeatedly—possibly until, or after, the onset of the Initial Upper Palaeolithic at least 45,000 years ago, when modern human mtDNA is first recorded in the sediments.

Denisova Cave consists of three chambers (designated Main, East and South Chambers) that contain deposits with stratigraphic sequences extending from the Middle Pleistocene to the Holocene epoch. The Pleistocene deposits have chronologies that have been constructed from the radiocarbon dating of bone, tooth and charcoal[5] (to around 50 thousand years ago (ka)) and optical dating of sediments[8] (to more than 300 ka). Optical ages for Main and East Chambers (Fig. 1a–c) can be aligned on a common time scale (Extended Data Fig. 1) but excavations are ongoing in South Chamber, where layers are only tentatively recognized. Mitochondrial DNA and nuclear DNA have been recovered from eight hominin fossils, enabling four to be assigned to Denisovans (Denisova 2, Denisova 3, Denisova 4 and Denisova 8)[1–4], three to Neanderthals (Denisova 5, Denisova 9 and Denisova 15)[5,6,12], and one to the child of a Neanderthal and a Denisovan (Denisova 11)[7]. However, there are too few fossils to enable the detailed reconstruction of the timing and sequence of hominin occupation, and the association of the early Middle Palaeolithic, middle Middle Palaeolithic and Initial Upper Palaeolithic assemblages identified at the site with specific hominin groups. Moreover, two Denisovan fossils (Denisova 3 and Denisova 4)—but no modern human remains—have been recovered

from the Initial Upper Palaeolithic layers, so it is debated whether archaic hominins or modern humans created the associated ornaments and bone tools[9–11].

A pilot study of DNA preservation in sediments from Denisova Cave identified ancient hominin mtDNA in 12 out of 52 samples[13], which suggested a path to reconstructing the occupational history of the site at higher resolution than is feasible from the scarce hominin fossil record. Here we report the analysis of 728 sediment samples, collected in a 10–15-cm grid-like pattern from the exposed Pleistocene deposits in all three chambers (Extended Data Figs. 2, 3a, b, Supplementary Information sections 1, 2). Using automated laboratory protocols, DNA was extracted from each sample, converted to single-stranded libraries and enriched for mammalian and hominin mtDNA[13,14], which we identified to the biological-family level using an established analysis pipeline[13].

## Patterns of DNA preservation

We identified ancient mammalian mtDNA in 685 samples (94%) from all sampled layers, including those older than 290 ka (Extended Data

[1]Max Planck Institute for Evolutionary Anthropology, Leipzig, Germany. [2]Centre for Archaeological Science, School of Earth, Atmospheric and Life Sciences, University of Wollongong, Wollongong, New South Wales, Australia. [3]Australian Research Council Centre of Excellence for Australian Biodiversity and Heritage, University of Wollongong, Wollongong, New South Wales, Australia. [4]Institute of Archaeology and Ethnography, Russian Academy of Sciences, Siberian Branch, Novosibirsk, Russia. [5]Department of Anatomy and Anthropology, Sackler Faculty of Medicine, Tel Aviv University, Tel Aviv, Israel. [6]Department of Human Molecular Genetics and Biochemistry, Sackler Faculty of Medicine, Tel Aviv University, Tel Aviv, Israel. [7]The Shmunis Family Anthropology Institute, The Dan David Center for Human Evolution and Biohistory Research, Tel Aviv University, Tel Aviv, Israel. ✉e-mail: elena_zavala@eva.mpg.de; zenobia@uow.edu.au; rgrob@uow.edu.au; mmeyer@eva.mpg.de

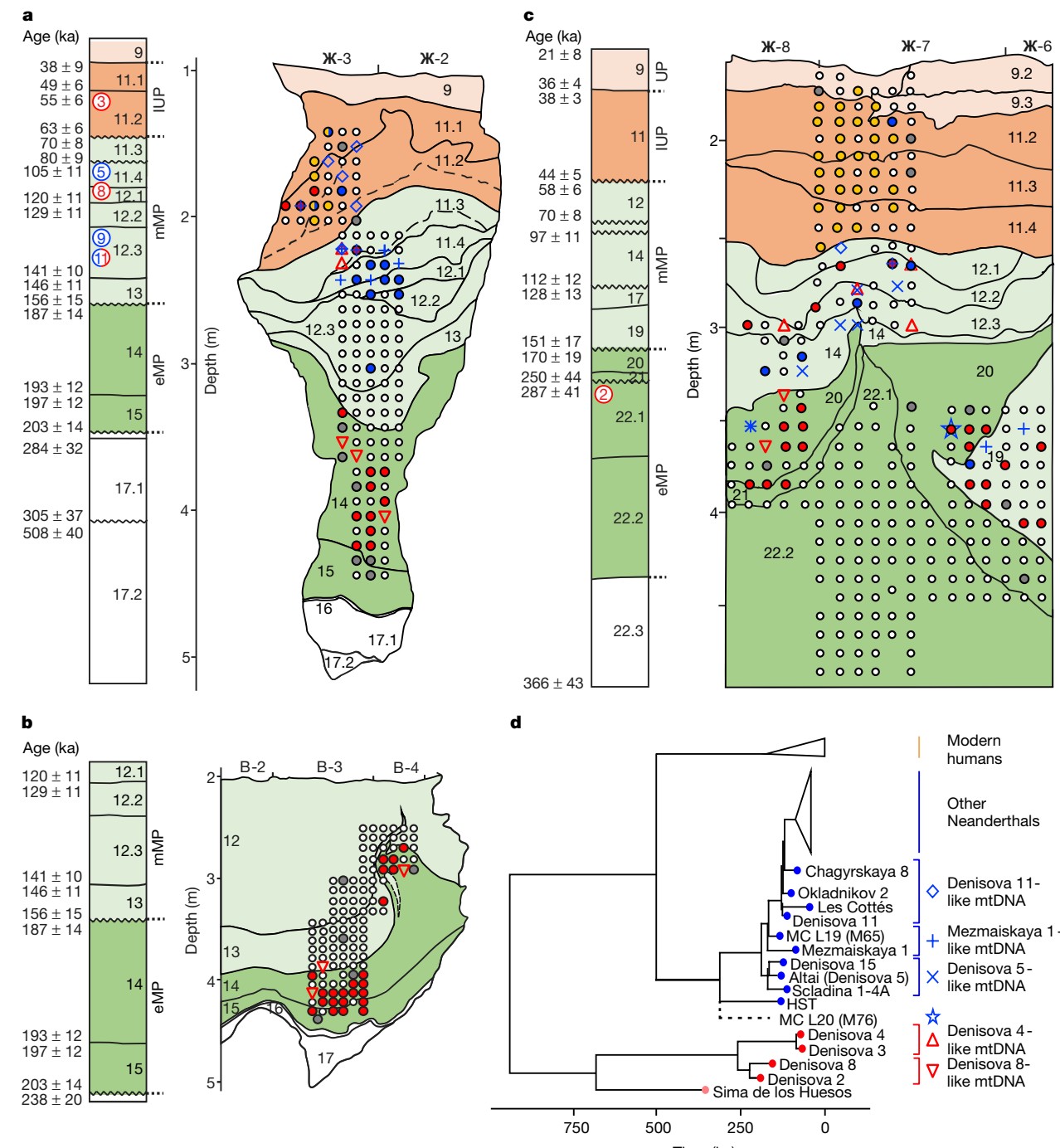

**Fig. 1 | Stratigraphic sequences in Denisova Cave, showing locations of sediment samples collected for mtDNA analysis and results obtained for ancient hominins. a**, East Chamber, southeast profile. **b**, East Chamber, northwest profile. **c**, Main Chamber, southeast profile. **d**, Phylogenetic tree of mtDNA genomes used as references to distinguish specific hominin lineages, and estimated placement of the Neanderthal mtDNA lineage identified in sample M76 from Main Chamber layer 20. Filled circles in **a**–**c** indicate the locations of individual sediment samples, and colours correspond to the hominin mtDNA detected: red (Denisovan), blue (Neanderthal), yellow (ancient modern human), grey (unidentified ancient hominin) and white (no ancient hominins detected). Other symbols denote samples for which mtDNA could be assigned to one of the specific hominin lineages in **d** (Denisovan, red open triangles; Neanderthal, blue open diamonds, crosses and star). Composite stratigraphic sections (modified from a previous publication[8]) to the left of each profile show modelled start and end ages (in ka) for sedimentary layers (uncertainties at 95.4% probability). Hominin specimen numbers are shown circled in the layer from which each fossil was recovered: Denisovan (red), Neanderthal (blue) and Neanderthal–Denisovan offspring (both colours)[5]. Dashed lines in profiles in **a** and **b** indicate areas in which layer assignment is uncertain[8]. Background shading denotes the associated archaeological assemblage: early Middle Palaeolithic (eMP) (dark green), middle Middle Palaeolithic (mMP) (light green), Initial Upper Palaeolithic (IUP) (dark orange) and Upper Palaeolithic (UP) (light orange).

Figs. 4, 5a, Supplementary Data 1). DNA retrieved from the deposits in all three chambers shows deamination-induced substitutions that are characteristic of ancient DNA[15,16]. These substitutions significantly increase with age (Extended Data Figs. 3d, 4a), which argues against extensive post-depositional leaching of DNA across layers[13,17]. We also observed a significant reduction in average DNA fragment length

and the number of mtDNA fragments recovered with increasing age (Extended Data Figs. 3e, f, 4b, c), although variability was greater across layers than for deamination, probably as a result of local differences in the geochemical environment. For example, the fewest ancient DNA fragments (none in some samples) were recovered from the youngest sampled unit (designated pdd-9) in South Chamber (Extended Data Fig. 3f), which is extensively phosphatized[8] and has slightly acidic pH values (between 6 and 6.5) (Supplementary Information section 3, Supplementary Data 2).

## Ancient hominin mtDNA

We detected ancient hominin mtDNA in 175 samples (24%), covering nearly all layers in all three chambers (Fig. 1a–c, Extended Data Figs. 3a, b, 5b). Four samples showed evidence for the presence of predominantly one haplotype and yielded sufficient mtDNA fragments to reconstruct mtDNA consensus sequences that are more than 80% complete (Supplementary Information sections 5, 6). Three of the sequences (samples E202 and E213 from East Chamber (layers 11.4 and 11.4/12.1) and sample M65 from Main Chamber (layer 19)) group with Neanderthals in phylogenetic trees built with previously published hominin mtDNAs (Supplementary Information section 7), specifically with Denisova 5, Denisova 15, Mezmaiskaya 1 and Scladina I-4A (Fig. 1d, Extended Data Fig. 6a). The fourth sequence (sample M71 from Main Chamber (layer 20)) is of the Denisovan type and falls basal to Denisova 2 and Denisova 8, albeit with low bootstrap support (Extended Data Fig. 6b). The most complete mtDNA sequence (over 99% of the genome reconstructed) for a Neanderthal from Main Chamber (M65) has a genetic age estimate of 140 ka (95.4% highest posterior density interval of 181–98 ka) (Supplementary Information section 7), consistent with the time of deposition of layer 19 (151 ± 17 to 128 ± 13 ka; here and below, uncertainties on optical ages are given at 95.4% probability)[8].

For the remaining 171 samples, we assigned mtDNA fragments to specific hominin groups by counting the number of fragments that support lineage-specific states at diagnostic sites that distinguish between modern human, Neanderthal and Denisovan mtDNA genomes. We distinguished three Neanderthal lineages: the Sima de los Huesos lineage (representing Neanderthals who lived approximately 430 ka in Spain and whose mtDNA is most closely related to that of Denisovans)[18,19]; the Hohlenstein–Stadel (HST) lineage, which falls basal to all other Neanderthal mtDNAs[20]; and the 'typical' Neanderthal mtDNA, known from all other Neanderthals. The presence of ancient modern human mtDNA was evaluated by restricting the analysis to deaminated fragments to mitigate the effect of present-day human DNA contamination. We identified Denisovan and typical Neanderthal mtDNA in 79 and 47 samples, respectively (based on 54–9,093 unique hominin mtDNA fragments), and modern human mtDNA in 35 samples (based on 55–2,200 deaminated fragments) (Fig. 1a–c, Extended Data Fig. 3a, b). We detected DNA from two hominin groups in ten samples, either within a single library or across libraries that were prepared from independent subsamples in some cases (Extended Data Fig. 6c, Supplementary Information section 4). In addition, we identified one sample (M76 from Main Chamber (layer 20)) containing hominin mtDNA fragments that support the branch shared by HST and typical Neanderthal mtDNA, but neither of the branches defining those lineages. This signal cannot be created by mixing mtDNA fragments from Neanderthals, Denisovans and ancient or present-day modern humans. On the basis of simulations with ancestralized Neanderthal mtDNA, the mtDNA in this sample is compatible with the presence of a previously unknown Neanderthal mtDNA lineage that diverged from typical Neanderthal mtDNA between 255 and 230 ka, 20 to 45 thousand years after the split of the HST and typical Neanderthal mtDNA lineages (Supplementary Information section 9).

The oldest hominin mtDNA recovered—identified as Denisovan—originates from a sample in Main Chamber layer 21, which began to accumulate 250 ± 44 ka. This provides the earliest genetic evidence for hominin occupation in Denisova Cave; Denisova 2 was found in layer 22.1, but is probably intrusive from an overlying layer and has an estimated age of 194–123 ka[5]. Among all 223 samples from the early Middle Palaeolithic layers in Main and East Chambers, 50 contained evidence for Denisovan mtDNA and only three (all from layer 20 in Main Chamber) for Neanderthal mtDNA. Two of these (M174 and M235) contain typical Neanderthal mtDNA and are from areas in which small-scale mixing with overlying sediments may have occurred[8]; the third (M76) is from the middle of the layer, and carries the previously unknown Neanderthal mtDNA lineage. These results point to Denisovans as the first and principal makers of the early Middle Palaeolithic assemblages, which are older than 170 ± 19 ka. Consistent with this interpretation, the detection of Neanderthal mtDNA in a sediment sample from early Middle Palaeolithic layer 14 in East Chamber in the pilot study[13] was due to an incorrect assignment, which was later corrected to middle Middle Palaeolithic layer 11.4 in this chamber[8] (Supplementary Information section 2). Our results also suggest that Neanderthals first occupied Denisova Cave towards the end of the early Middle Palaeolithic and may therefore have contributed to the production of these assemblages in their later stages.

Forty out of 173 samples from the middle Middle Palaeolithic layers in Main and East Chambers (deposited approximately 160–60 ka) yielded Neanderthal and/or Denisovan mtDNA, with both present in six samples (Fig. 1a–c). DNA from both groups also occurs in the deformed Middle Palaeolithic layers in South Chamber (Extended Data Fig. 3b). Notably, sediments deposited between 120 ± 11 and 97 ± 11 ka in Main and East Chambers produced no traces of Denisovan mtDNA, whereas 12 samples contained Neanderthal mtDNA. This suggests that only Neanderthals may have occupied the cave during that period, and possibly for most of Marine Isotope Stage (MIS) 5 (Fig. 2).

Only ancient modern human mtDNA was detected in the Initial Upper Palaeolithic and Upper Palaeolithic layers in Main Chamber (layers 11.4 and above, deposited 44 ± 5 to 21 ± 8 ka), except for one sample from Initial Upper Palaeolithic layer 11.2 that yielded Neanderthal mtDNA (Fig. 1c). The association between Initial Upper Palaeolithic assemblages and the appearance of modern humans is further supported by the recovery of modern human mtDNA from a sample from Initial Upper Palaeolithic layer 11 in South Chamber, which was deposited after 47 ± 8 ka (Extended Data Fig. 3a). The situation in East Chamber is more complex: Denisovan, Neanderthal and ancient modern human mtDNAs were recovered from Initial Upper Palaeolithic layer 11.2, and Neanderthal and ancient modern human mtDNA from Initial Upper Palaeolithic layer 11.1 (Fig. 1a). Given these results and the recovery of two Denisovan fossils (Denisova 3 and Denisova 4) from layers associated with Initial Upper Palaeolithic assemblages, we cannot discount the possibility that—in addition to modern humans—Denisovans and Neanderthals may have been present during the period of Initial Upper Palaeolithic production[9–11].

For 34 out of 37 samples that yielded 100 or more deaminated hominin mtDNA fragments, we identified similarities with specific Neanderthal and Denisovan mtDNA genomes using a $k$-mer-based approach[21,22] (denoted by symbols other than circles in Fig. 1a–c, Extended Data Fig. 3b, Supplementary Information section 8). This analysis revealed that the early Middle Palaeolithic layers in Main and East Chambers and the earliest middle Middle Palaeolithic layer in East Chamber, which span the period between 250 ± 44 and 146 ± 11 ka, contain Denisova-2- and Denisova-8-like mtDNA fragments. This contrasts with the Denisovan mtDNA recovered from middle Middle Palaeolithic layers deposited after 80 ± 9 ka, which yielded assignments to Denisova-3- and Denisova-4-like sequences, as did a sample from South Chamber. These results suggest a transition in mtDNA sequences sometime in the 146–80-ka interval, possibly reflecting different Denisovan populations. Our results also align well with the modelled ages of Denisova 2, Denisova 3 and Denisova 4, and with the relative age of Denisova

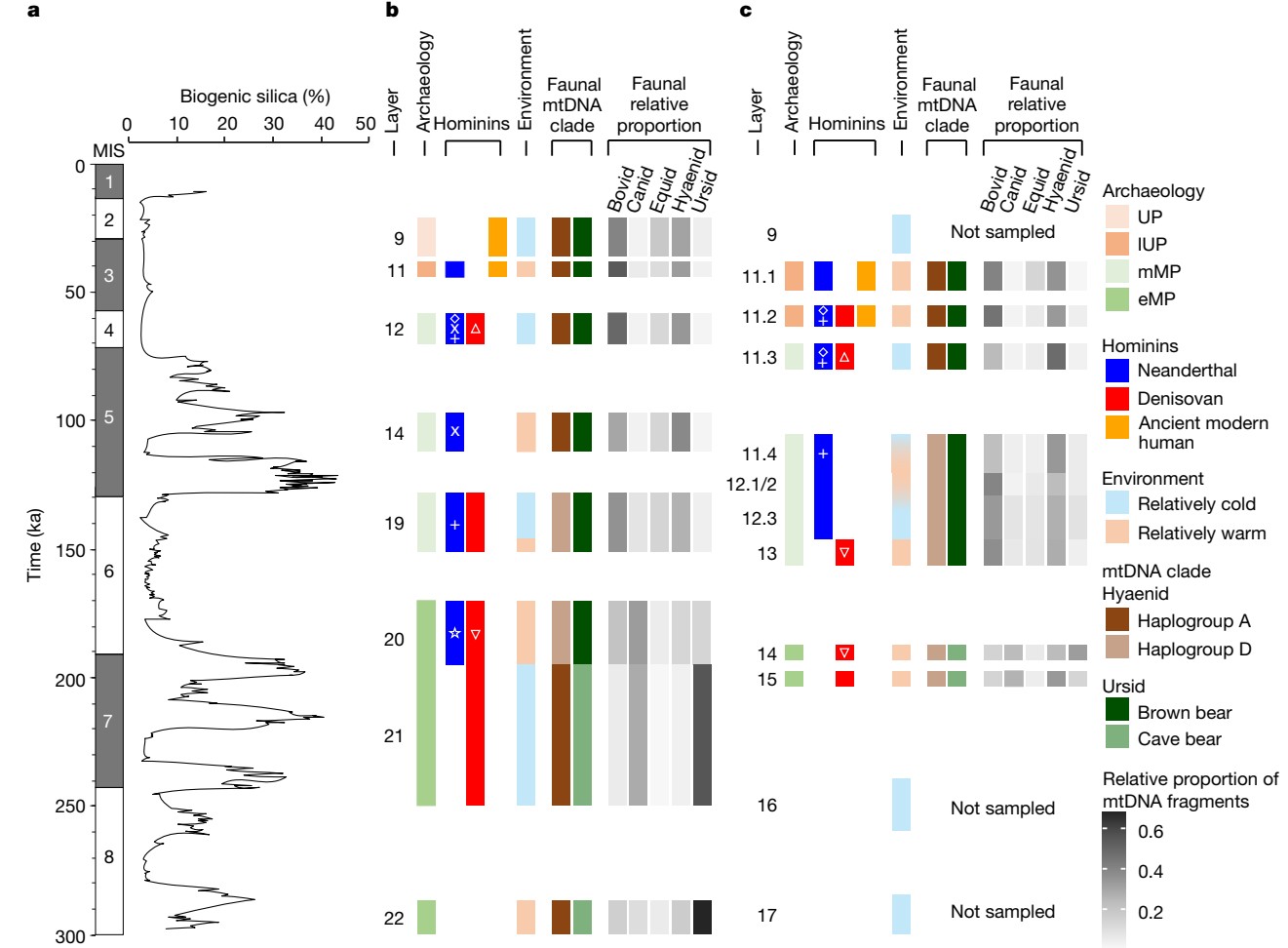

**Fig. 2 | Timeline of hominin and faunal mtDNA presence, archaeological phases and environmental records at Denisova Cave. a**, Baikal Drilling Project 1996 composite biogenic silica record of diatom productivity in Lake Baikal, a proxy for regional annual temperature[28]. **b**, **c**, Summary data for Main Chamber (**b**) and East Chamber (**c**). Start and end ages for layers and corresponding data are based on the common time scale in Extended Data Fig. 1. Time intervals in white represent gaps in the stratigraphic sequence, or were not sampled. Archaeological phases (early Middle Palaeolithic, middle Middle Palaeolithic, Initial Upper Palaeolithic and Upper Palaeolithic) follow the colour scheme in Fig. 1 and Extended Data Fig. 1. Genetic data for hominins are from Fig. 1 (excluding samples M174 and M235 from Main Chamber, which are thought to be out of context), and for dominant hyaenid and ursid populations from Extended Data Fig. 10. Symbols for specific hominin mtDNA lineages (Fig. 1d) are inset in white. Environmental conditions are inferred from pollen records and skeletal remains of vertebrate fauna[8]. Relative proportions of bovid, canid, equid, hyaenid and ursid mtDNA are from Extended Data Figs. 8, 9.

8 inferred from molecular dating[5]. Sediments deposited between about 130 and 100 ka (and possibly longer, given the subsequent time gap of 20 millennia)—during MIS 5—contain mtDNA and fossil evidence only of Neanderthals, with Denisova-11-like mtDNA sequences only appearing in sediments deposited after 80 ka.

## Ancient faunal mtDNA

All large mammals present in the palaeontological record of Denisova Cave (currently available for Main and East Chambers only[8]) were also identified in the sediment DNA (Extended Data Fig. 7a, Supplementary Information section 10). In addition, ancient mtDNA from camelids was found in one sample from East Chamber (layer 12), consistent with the Pleistocene presence of *Camelus knoblochi* in the region[23]. In contrast to large mammals, small mammals (such as Spalacidae, Leporidae and Sciuridae) are largely absent from the genetic data (Extended Data Fig. 7b), which may be due to their lower biomass or underrepresentation among the capture probes. The sharp boundaries between the faunal mtDNA composition in some of the adjacent layers (for example, between layers 22.1 and 21/20 in Main Chamber) (Extended Data Fig. 8)

provide additional evidence that post-depositional leaching of DNA is limited, if it occurs.

Despite the highly fragmentary nature of fossil remains at Denisova Cave[8] and the varying quantities of DNA that may be deposited by different species, changes in the relative abundance of mammalian DNA over time are broadly consistent with changes in the skeletal records for some families, such as bovids, hyaenids, ursids and canids (Extended Data Fig. 7c, d). Genetic data also provide the opportunity to study faunal diversity at the species or population level, in cases in which comprehensive reference data are available (as is the case for elephantids, ursids and hyaenids) (Supplementary Information section 11). Elephantid mtDNA was assigned predominantly to woolly mammoth in all layers, whereas the relative abundance of ursid species shifted from predominantly cave bear mtDNA in layers deposited before 187 ± 14 ka to exclusively brown bear mtDNA after 112 ± 12 ka (Extended Data Fig. 10). We also detected the presence of three previously described mtDNA haplogroups of the genus *Crocuta* (spotted and cave hyaenas)[24]. Layers deposited before 200 ka and after 120–80 ka contain mostly mtDNAs seen in African spotted hyaenas and European cave hyaenas (haplogroup A), whereas layers of intermediate age contain mtDNA

predominantly from east Asian cave hyaenas (haplogroup D) and some from European cave hyaenas (haplogroup B) (Extended Data Fig. 10). The Altai Mountains may therefore have been a contact zone for both hominins and distinct lineages of hyaena and other fauna, as previously suggested by studies of mammalian skeletal remains[25].

At least two major turnovers of large mammals are evident from the sediment DNA (Fig. 2). First, marked changes in the relative proportions of mtDNA fragments of bovids, canids, equids, hyaenids and ursids, a turnover in hyaena mtDNA haplogroups and a shift from cave bears to brown bears occurred about 190 ka, contemporaneous with the climatic transition from an interglacial period (MIS 7) to a glacial period (MIS 6). The earliest traces of Neanderthal mtDNA also appear around this time. A second turnover took place between about 130 and 100 (or 80) ka, during and after the climatic transition from MIS 6 to MIS 5: mtDNA proportions of bovids, canids, felids and ursids declined, whereas those of cervids and equids increased, and cave bears and two hyaena haplogroups disappeared (Extended Data Fig. 10). This period is notable also for the absence of Denisovan mtDNA in the cave sediments. These changes suggest that turnovers in hominin and faunal populations may have been linked, and related to ecological factors[25].

## Discussion

The identification of archaic hominin mtDNA in 175 sediment samples exceeds by an order of magnitude the number of hominin fossils retrieved from the deposits in Denisova Cave and provides a genetic profile of the presence of hominins in nearly all of the Pleistocene layers (Fig. 2). These data are complemented by faunal mtDNA sequences from 685 samples, which provide information about the diversity of other large mammals and changes in their relative abundance. However, we caution that the inferred sequence of hominin and faunal occupation is constrained by several factors: the existence of two major gaps in the stratigraphic record (170–156 and 97–80 ka), the time-averaging inherent in the accumulation of each sediment layer, the post-depositional disturbance of some layers owing to burrowing animals or small-scale mixing[8,26], and the precision of the optical ages used to construct the site chronology.

Beyond reconstructing the occupational history of Denisova Cave, our results have wider implications for understanding the human past. First, fragments of Denisovan mtDNA recovered from the middle Middle Palaeolithic layers deposited in Denisova Cave after 80 ka consistently show the highest similarity to the mtDNA of Denisova 3 and Denisova 4, as do mtDNA fragments retrieved from sediments at Baishiya Karst Cave on the northeastern flank of the Tibetan Plateau that are broadly contemporaneous in age (70–45 ka)[27]. This pattern suggests that this lineage was the most abundant mtDNA type carried by Denisovans after 80 ka. Palaeontological studies[25] have suggested that Pleistocene mammals migrated from southeast Asia, along the eastern foothills of the Himalayas, to the northwest Altai. These faunal migrations may have spurred the dispersal of Denisovans into the region in which their remains were first discovered. Second, the presence of Neanderthal mtDNA before 170 ka further constrains the timing of an early event in Neanderthal history—the replacement of the mtDNA lineage found in Neanderthals who lived 430 ka in Spain by mtDNA that introgressed from early ancestors of modern humans[18,19]—to between 430 and 170 ka. High-resolution profiling of sediment DNA can therefore provide an effective means of filling gaps in our knowledge of human evolutionary history and palaeoecology, independent of the discovery of skeletal remains.

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

## Methods

No statistical methods were used to predetermine sample size. The experiments were not randomized, and investigators were not blinded to allocation during experiments and outcome assessment.

### Collection of sediment samples

We collected a total of 728 sediment samples in a 10–15-cm grid-like pattern from the exposed Pleistocene deposits in Main ($n = 274$), East ($n = 252$) and South ($n = 202$) Chambers (Extended Data Figs. 2b–d, 3a, b). We deviated from this pattern only if rocks were in the way or if a sample was clearly located on the boundary between layers. Gloves, face masks and hair nets were worn to minimize contamination by modern DNA. At the time of sample collection, a small area of the exposed profile at each sample location was first cleaned back to a depth of approximately 1 cm using a sterilized scalpel blade. Each sample was then collected using a new scalpel blade, which was inserted into the deposit, and the sediment extracted was carefully placed in a separate zip-lock plastic bag for each sample. This process was repeated at each location until sufficient sediment (several grams) had been collected. Each zip-lock bag was sealed immediately with duct tape and labelled with its sample number and prefix M, E or S to denote Main, East or South Chamber, respectively. Sample positions and corresponding layers were recorded in a field notebook, and layer assignments were checked for accuracy by M.B.K. in the field and also against high-resolution photographs. Sample numbers and locations are displayed in Supplementary Figs. 1–3. All materials were acquired as part of an agreement of scientific cooperation between the Institute of Archaeology and Ethnography, Siberian Branch of the Russian Academy of Sciences and the Max Planck Institute for Evolutionary Anthropology for projects in the field of palaeogenetics in North Asia, signed on 25 December 2018 and valid for a duration of five years. The Institute of Archaeology and Ethnography, Siberian Branch of the Russian Academy of Sciences oversees the excavations at Denisova Cave and obtained all of the permits necessary for conducting archaeological fieldwork and research associated with this project from the Ministry of Culture of the Russian Federation.

### Common time scale for Main and East Chambers

We constructed a common time scale for the Pleistocene stratigraphic sequences in Main and East Chambers, but excluded South Chamber because of various stratigraphic complications that preclude definitive layer assignments to most of the Pleistocene deposit exposed at the time of sample collection (Supplementary Information section 2).

The stratified sequences in each chamber are numbered by layer, but layers with the same number cannot be traced stratigraphically between the chambers and are not necessarily equivalent in age. To correlate stratigraphic layers between chambers, a previous study[8] constructed a separate Bayesian statistical age model for each chamber using their large dataset of optical ages, and then established isochrons (lines of equal age) between the chambers, using the modelled start and end ages for each depositional phase (figure 3 and extended data table 1 in ref. [8]). Bayesian age models were constructed separately for each chamber because there is no a priori reason to assume that sediments have accumulated continuously, or at the same rate, in each of the chambers. Time gaps in the stratigraphic sequence may therefore differ between the chambers owing to erosional events or periods of little or no sediment deposition.

Here we created a common time scale for the Pleistocene stratigraphic sequences in Main and East Chambers, using the modelled start and end ages for each depositional phase and the modelled time gaps (Extended Data Fig. 1). To construct this time scale, only point estimates of modelled age and not their associated uncertainties were taken into consideration; for example, if the start and end ages for a specific phase are $100 \pm 10$ and $50 \pm 5$ ka, respectively, then the corresponding time depth (size of coloured box in Extended Data Fig. 1a, b) is estimated to extend from 100 to 50 ka. The modelled ages for Main and East Chambers were projected horizontally onto a linear scale to derive a common time scale for the depositional phases in these two chambers (Extended Data Fig. 1c). We use the common timeline to display the mtDNA data and to show (Extended Data Fig. 1d) the Bayesian modelled ages (95.4% highest posterior density interval) for individual hominin fossils[5].

### Sampling, DNA extraction, library preparation and shotgun sequencing

Sampling, DNA extraction and library preparation were performed in a dedicated clean room at the Max Planck Institute for Evolutionary Anthropology (Leipzig). Sediment subsamples of between 29 and 191 mg were transferred into 2-ml tubes using antistatic spatulas. Before use, each spatula was soaked in 12% bleach for approximately 15 min with occasional mixing, washed thoroughly with water, dried and UV-treated with a dose of 7 J cm$^{-2}$ in a UV-C crosslinker from both sides. One ml of extraction buffer[29] was added to subsamples with less than 100 mg of sediment, and 2 ml to samples with 100 mg or more. The resulting lysates were incubated overnight at 37 °C on rotation. Aliquots of 150 µl of lysate were purified on a BRAVO NGS workstation B (Agilent Technologies) following a previously described bead-based protocol[29] using binding buffer 'D'. The remaining lysate was stored at −20 °C. Negative controls containing no sample material were included for both the lysis and purification steps. Supplementary Data 1 provides sample information.

The entire volume of DNA extract (30 µl) from each sample was converted into a single-stranded DNA library using the BRAVO NGS workstation B as previously described[30]. Approximately six million molecules of a control oligonucleotide were added into each sample during the library preparation to evaluate the potential presence of inhibitors, which may reduce the efficiency of the process[31]. The number of sample and control library molecules obtained in each reaction was determined using two real-time PCR assays[30]. All libraries were tagged with two indices, amplified to PCR plateau, and purified as described in the aforementioned protocol[30].

Aliquots of 1 µl from each library were pooled into sets of up to 92 samples (including controls) and sequenced on an Illumina MiSeq v3 platform in 76 cycle paired-end reads using micro or nano flowcells. Base calling was completed using Bustard (Illumina). The resulting shotgun data were evaluated only for the presence of the expected index pairs, which was confirmed in all sequencing runs.

### Hybridization capture for mammalian and hominin mtDNA

Each library was enriched separately with two mtDNA probe sets, one targeting 242 mammalian mtDNA genomes[13] and the other using the revised Cambridge Reference Sequences (rCRS[32]) for targeting hominin mtDNA, in two successive rounds of on-bead hybridization capture following a previously published protocol[13] with minor adjustments. All capture reactions were performed in sets of 384 samples (including extraction and library negative controls), using the BRAVO NGS workstation B. For some samples, 30 PCR cycles were performed in post-capture amplification of the enriched libraries (thus reaching PCR plateau for all libraries). For others, PCR cycles were reduced to between 12 and 16.

Captured libraries were pooled by combining 5-µl aliquots from each library in sets of approximately 92 libraries (including controls) for mammalian mtDNA enriched libraries, and sets of approximately 180 libraries (including controls) for human mtDNA enriched libraries. Each pool was sequenced on the Illumina HiSeq 2500 platform. Base calling was completed using Bustard (Illumina). The resulting capture data were evaluated as described in Supplementary Information sections 3, 4 for the presence of ancient mammalian and/or hominin DNA.

## Identification of mammalian taxa

The initial processing of the mammalian capture data was performed using a previously described pipeline[13]. In brief, overlapping paired-end reads were merged into full-length molecule sequences using leeHom[33] (https://bioinf.eva.mpg.de/) and mapped to the 242 mammalian mitochondrial genomes included in the capture probe design[13]. Reads that could not be overlap-merged, unmapped sequences and sequences shorter than 35 base pairs (bp) were removed. Only a single sequence was retained from duplicates showing perfect sequence identity, and sequences with fewer than two duplicates were removed. All unique sequences were then assigned to mammalian taxa at the family level using BLAST[34] and the lowest common ancestor algorithm implemented in MEGAN[35]. Sequences assigned to each family were mapped to the mitochondrial genomes of all species available within each family (Supplementary Data 3), requiring a mapping quality score of at least 25, and residual duplicate sequences were removed using bam-rmdup (https://github.com/mpieva/biohazard-tools) based on the sharing of identical alignment start and end coordinates. The mtDNA genome that produced the largest number of aligned sequences after removal of PCR duplicates was then used to determine the number of DNA fragments assigned to the respective family, as well as the frequency of cytosine (C) to thymine (T) substitutions at fragment ends.

Identification of 'ancient taxa' (that is, taxa for which ancient DNA sequences were retrieved) was performed according to the following criteria: (1) number of fragments assigned to a given taxon had to comprise at least 1% of the total number of taxonomically identified fragments; (2) at least 10 putatively deaminated fragments (that is, fragments showing C-to-T substitutions at the 5′ and/or 3′ terminal bases) had to be present; (3) frequency of C-to-T substitutions was required to be significantly higher than 10% based on 95% binomial confidence intervals at one or both termini; and (4) the fragments had to yield a coverage of at least 500 bp of the mtDNA reference genome. Supplementary Data 1 provides information on each sample and controls. Statistical testing was performed using R version 3.5.1.

## Identification of ancient hominin DNA fragments

The processing of the hominin mtDNA capture data was performed as previously described[13], using an analysis pipeline that differed from the processing of the mammalian mtDNA capture data (Supplementary Information section 3) only in that full-length molecule sequences were mapped to the rCRS of the human mitochondrial genome, and duplicates were removed based on alignment start and end coordinates using bam-rmdup (https://github.com/mpieva/biohazard-tools) before sequences were assigned to mammalian taxa.

To ensure that all libraries were sequenced to a sufficient depth, libraries that produced less than three sequence duplicates, on average, were sequenced deeper, and the final data were merged for analysis. The presence of ancient hominin DNA was determined according to the following criteria: (1) number of fragments assigned to hominins had to comprise at least 1% of the total number of taxonomically identified fragments; (2) at least 10 putatively deaminated fragments (that is, fragments showing a C-to-T substitution at the three 5′ and/or 3′ terminal bases) had to be present; and (3) frequency of C-to-T substitutions was required to be significantly higher than 10% based on 95% binomial confidence intervals at both termini. For some samples, multiple subsamples were taken and from some lysates, multiple libraries were prepared. The resulting data were analysed on a per-library basis, as well as merged by lysate (subsample). Supplementary Data 1 provides results of individual and merged data, and information on each sample and control. Statistical testing was performed using R version 3.5.1.

## Reporting summary

Further information on research design is available in the Nature Research Reporting Summary linked to this paper.

## Data availability

The mitochondrial consensus sequences reported from Main Chamber layers 19 (M65) and 20 (M71), and from East Chamber layers 11.4 (E202) and 11.4/12.1 (E213), are available in the Dryad digital repository (https://doi.org/10.5061/dryad.k3j9kd567), and the raw data for each mammalian mtDNA and human mtDNA enriched library are available in the European Nucleotide Archive under accession number PRJEB44036. Any other relevant data are available from the corresponding authors upon reasonable request.

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

**Acknowledgements** This project was funded by the Max Planck Society, the European Research Council (grant agreement no. 694707 to S.P.) and the Australian Research Council (fellowships FT150100138 to Z.J., FT140100384 to B.L. and FL130100116 to R.G.R.). K.O. was supported by an Australian Government Research Training Program Award. M.V.S., M.B.K. and A.P.D. were supported by RFBR, project no. 20-29-01011. V.S. acknowledges funding from the Alon Fellowship. We thank S. Grote, L. Jauregui, S. Lin, F. Mafessoni and L. Skov for help with the organization of laboratory work and data visualization; J. Lihanova, F. Müller, B. Schellbach and A. Weihmann for help with sample preparation and sequencing; and V. Vaneev for field assistance.

**Author contributions** E.I.Z., Z.J., S.P., R.G.R. and M.M. designed the study. Z.J., B.L., K.O. and R.G.R. collected samples in the field. M.V.S., M.B.K., A.P.D., Z.J. and R.G.R. provided archaeological, stratigraphical and geochronological context and interpretation. E.I.Z., E.E., S.N., J.R. and A.S. performed laboratory experiments. E.I.Z., Z.J., B.V., C.d.F., F.R., V.S. and J.K. performed, aided in or supervised data analysis and visualization. E.I.Z., Z.J., R.G.R. and M.M. wrote the manuscript with input from all authors.

**Funding** Open access funding provided by Max Planck Society.

**Competing interests** The authors declare no competing interests.

**Additional information**
**Correspondence and requests for materials** should be addressed to E.I.Z., Z.J., R.G.R. or M.M.

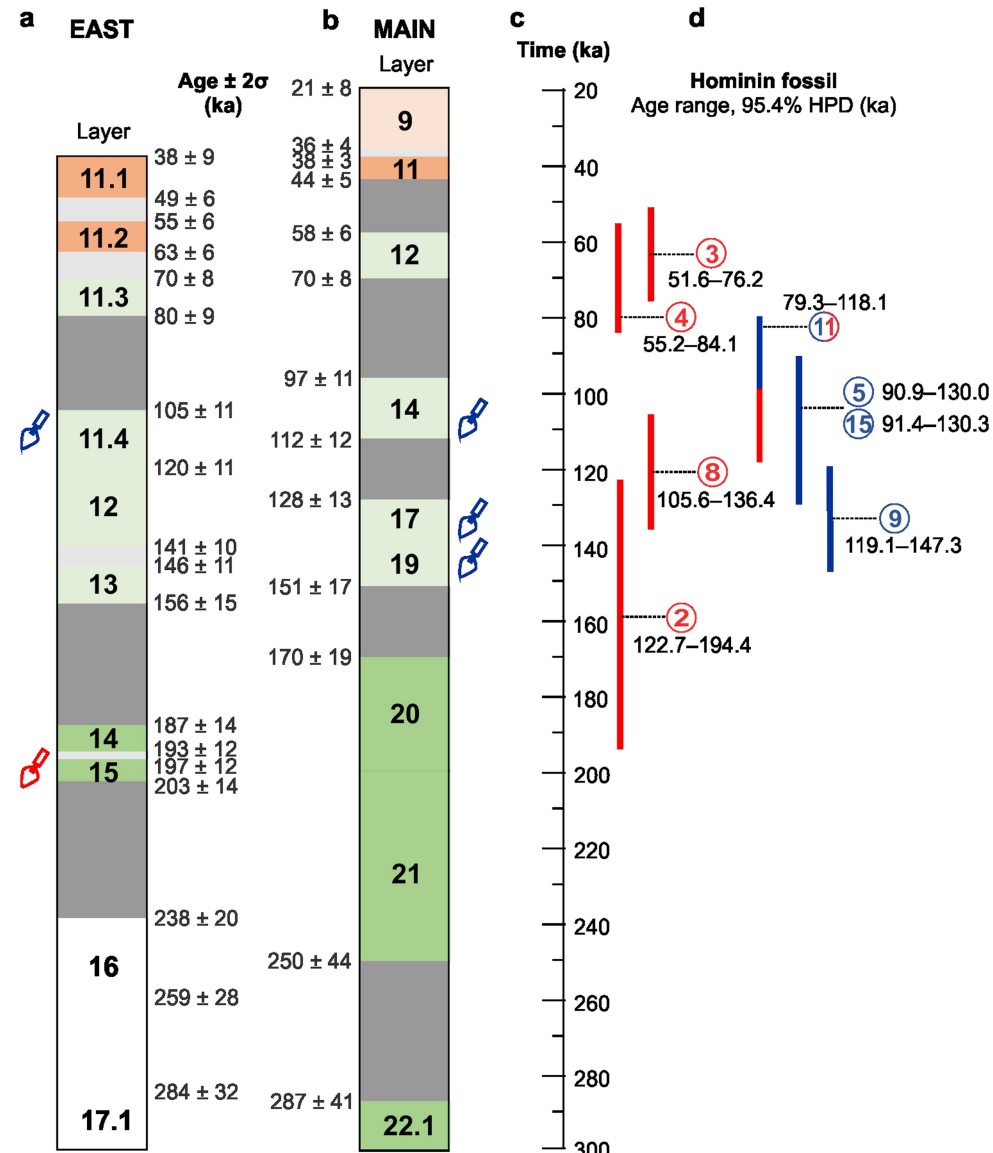

**Extended Data Fig. 1 | Chronologies for sedimentary layers and hominin fossils. a, b,** Schematic stratigraphic sequences for East Chamber (**a**) and Main Chamber (**b**), with ages and layer assignments from a previous publication[8]. Archaeological associations for each layer are colour-coded to differentiate between the four main artefact phases (early Middle Palaeolithic, dark green; middle Middle Palaeolithic, light green; Initial Upper Palaeolithic, dark orange; Upper Palaeolithic, light orange). White areas indicate layers with no archaeology, dark grey areas denote modelled time gaps and light grey areas denote time gaps between point estimates of age for successive layers that have overlapping age uncertainties. Trowel symbols indicate the layers from which sediment-derived Denisovan (red) and Neanderthal (blue) mtDNAs were recovered in a pilot study[13] (corrected for the misattribution of a sample in East Chamber to layer 14 instead of layer 11.4). **c,** Common time scale constructed for Main and East Chambers (Methods) to facilitate comparison between layers and display mtDNA data. **d,** Age ranges (95.4% highest posterior density (HPD)) for fossils of Neanderthals (blue), Denisovans (red) and the Neanderthal–Denisovan offspring (Denisova 11, both colours), plotted on the common time scale. Fossil ages were estimated using a Bayesian model incorporating radiocarbon, optical and uranium-series ages, stratigraphic information and genetic data generated from hominin DNA sequences[5].

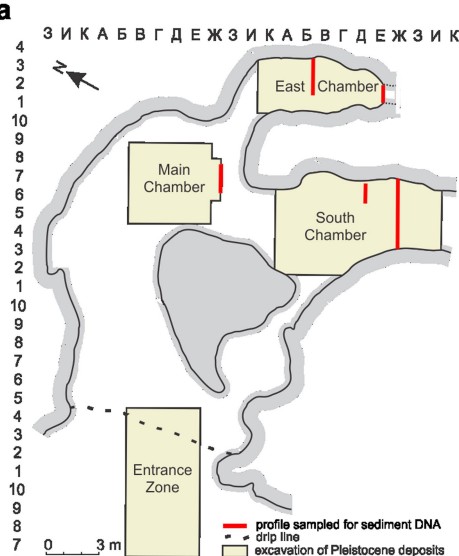

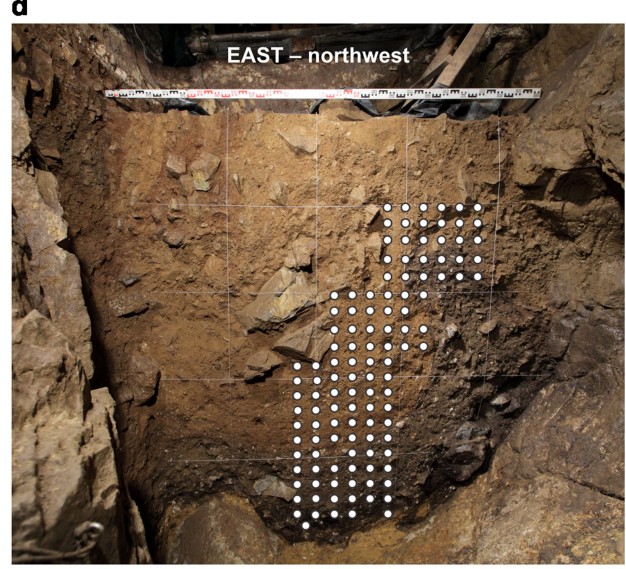

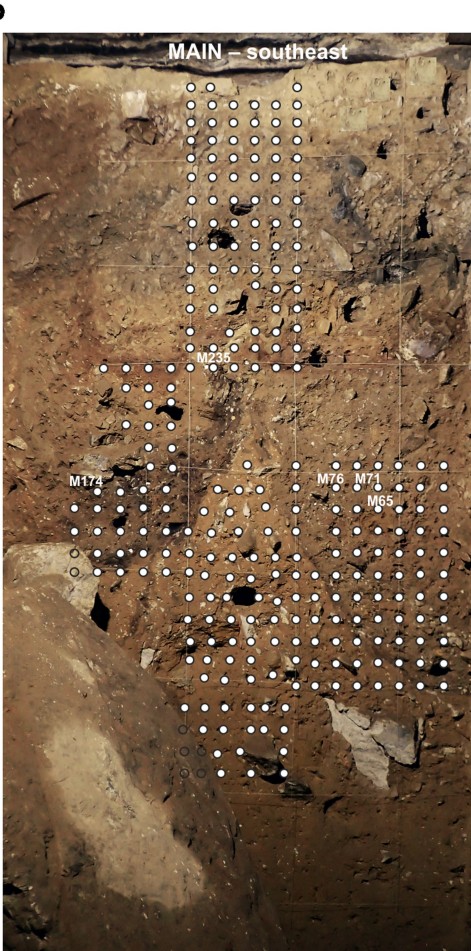

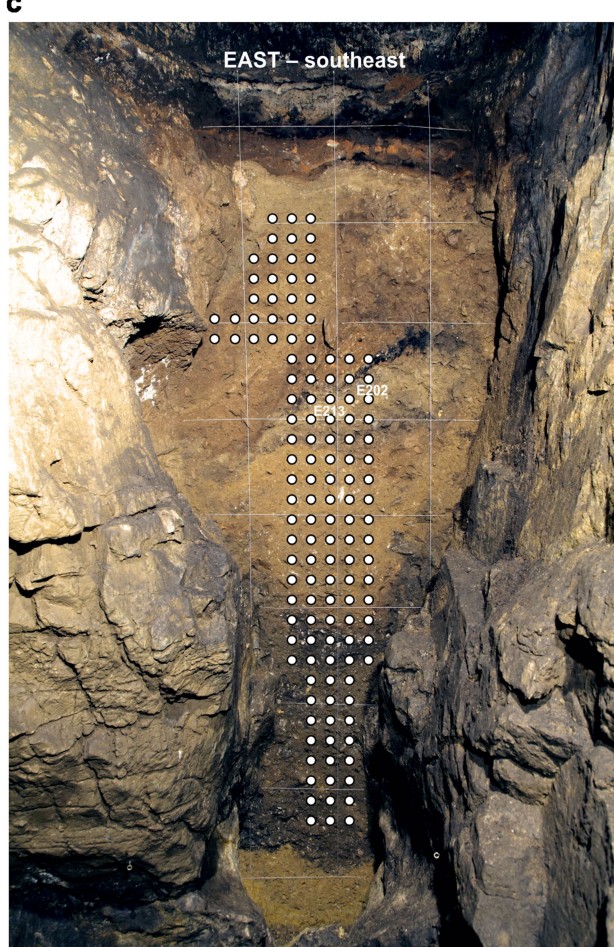

**Extended Data Fig. 2 | Locations of sediment DNA samples. a**, Plan of cave interior, showing location of each chamber. Red lines denote stratigraphic profiles sampled in 2017. Grid coordinates for the excavation squares are shown along the top and left side of the plan, and the corresponding squares (each consisting of a Cyrillic letter and a number) are shown at the top of the stratigraphic profiles in Fig. 1a–c, Extended Data Figs. 3a, b, 5, 8, 9, Supplementary Figs. 1, 2, 3b, d. **b**, Southeast profile of Main Chamber after excavations in 2016. **c**, Southeast profile of East Chamber after excavation in 2015. **d**, Northwest profile of East Chamber after excavation in 2016. White circles represent samples collected for sediment DNA analysis. Individual sample numbers are provided in Supplementary Figs. 1 (Main) and 2 (East); sample numbers referred to in the text are indicated in **b**, **c**. Sample locations in South Chamber are indicated in Extended Data Fig. 3a, b, Supplementary Fig. 3.

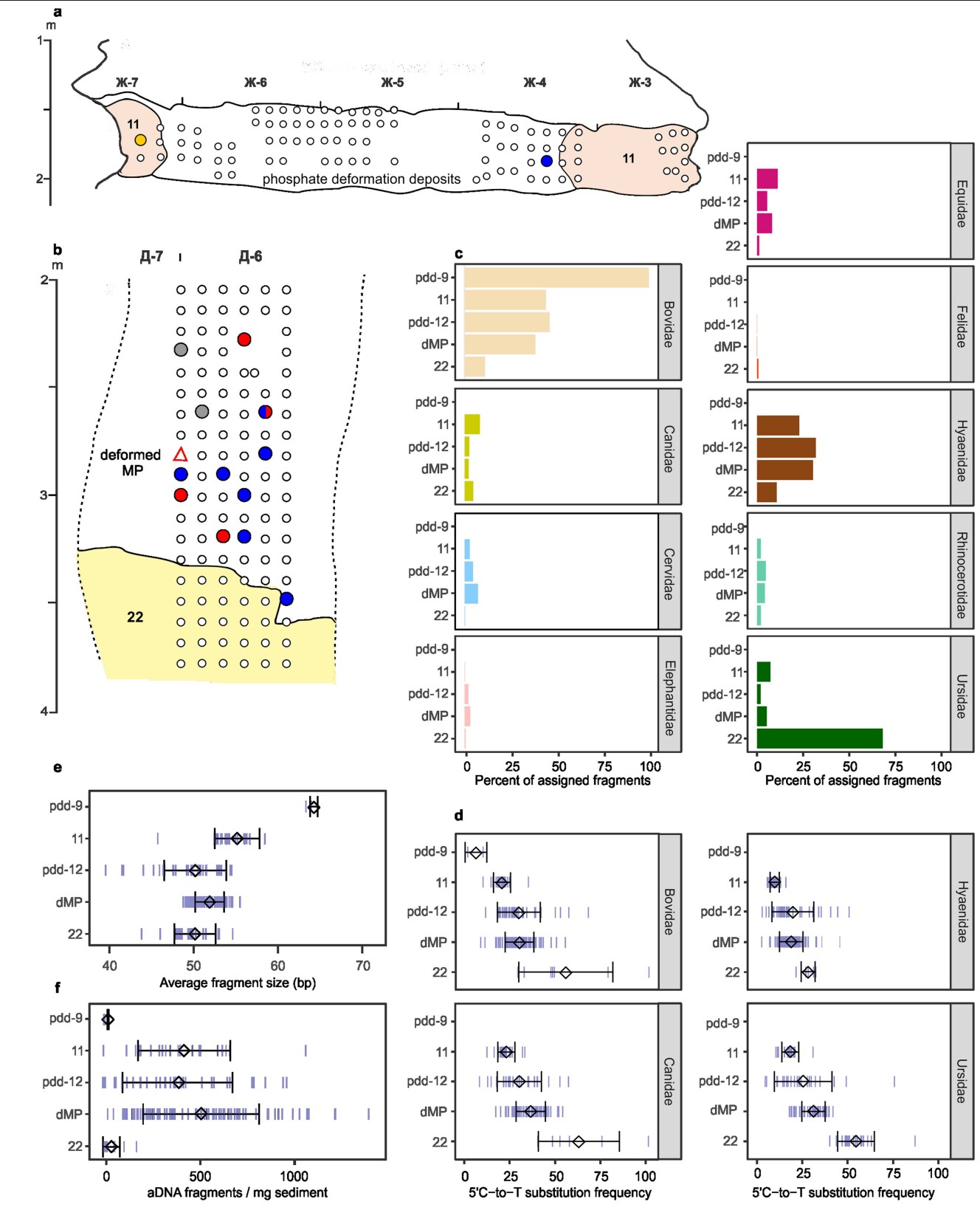

**Extended Data Fig. 3** | See next page for caption.

**Extended Data Fig. 3 | Ancient hominin and mammalian mtDNA data for sediment samples in South Chamber. a**, **b**, Sample locations in upper section (**a**) and lower section (**b**) of southeast profile, using the same symbols for hominins as in Fig. 1. Layer 11 (associated with the Initial Upper Palaeolithic) and layer 22 are numbered, but other layers are denoted as either deformed Middle Palaeolithic (dMP) or as phosphate deformation deposits (pdd-9 and -12). The dMP and pdd deposits have been substantially affected by post-depositional deformation and phosphatization, respectively (Supplementary Information section 2). **c**, Proportion of mtDNA fragments assigned to different families of large mammal in each layer, arranged in relative stratigraphic order. Values were obtained by averaging across the percentages of fragments assigned to each family in all samples from a specified layer. **d**, The 5′ C-to-T substitution frequencies (putative deamination rates, in per cent) of mtDNA fragments assigned to bovid, canid, hyaenid and ursid in each layer. Individual values are shown as vertical bars, and the mean and s.d. values by black symbols. **e**, Average size (base pairs (bp)) of mammalian mtDNA fragments in each layer; individual values are shown as vertical bars and the mean and s.d. values by black symbols. **f**, Number of unique mtDNA fragments in each layer assigned to mammalian taxa per milligram of sediment from each library; individual values are shown as vertical bars and the mean and s.d. values by black symbols. Spearman's correlation test (one-sided) was used to test for correlation between stratigraphic depth (layer) and 5′ C-to-T substitution frequency (minimum $n = 112$, positive correlation, maximum $P = 1.8 \times 10^{-5}$), number of unique ancient mtDNA fragments (minimum $n = 79$, negative correlation, maximum $P = 1.7 \times 10^{-5}$), and average fragment size (minimum $n = 79$, negative correlation, maximum $P = 1.8 \times 10^{-8}$).

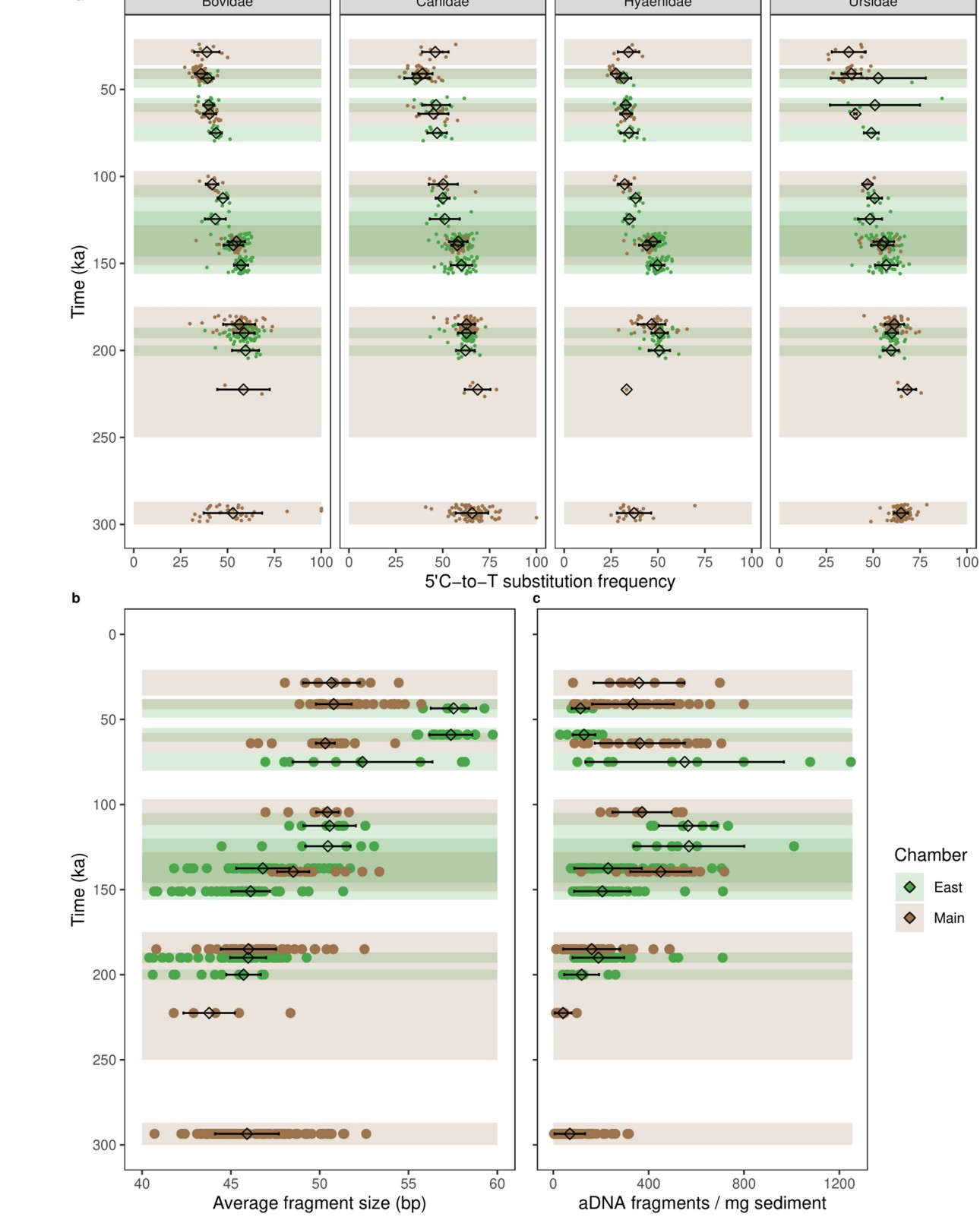

**Extended Data Fig. 4** | See next page for caption.

**Extended Data Fig. 4 | Ancient mammalian mtDNA preservation in Main and East Chambers. a**, The 5′ C-to-T substitution frequencies (putative deamination rates, in per cent) of mtDNA fragments assigned to bovid, canid, hyaenid and ursid, plotted as a function of time. **b**, **c**, Average size of mammalian mtDNA fragments (**b**) and number of unique mtDNA fragments assigned to taxa per milligram of sediment from each library (**c**), plotted as a function of time. Individual values for Main and East Chambers are shown as brown and green circles, respectively, and the mean and s.d. values by black symbols. The latter symbols are positioned on the time axis at the mean age for the relevant layer, and the time intervals for the corresponding layers in Main and East Chambers are denoted by brown and green shading, respectively (Extended Data Fig. 1). Spearman's correlation test (one-sided) was used to test for correlation between stratigraphic depth (layer) and 5′ C-to-T substitution frequency (minimum $n = 133$, positive correlation, maximum $P = 2.7 \times 10^{-8}$), number of unique ancient mtDNA fragments (minimum $n = 222$, negative correlation, maximum $P = 0.02$), and average fragment size (minimum $n = 222$, negative correlation, maximum $P = 2.0 \times 10^{-25}$).

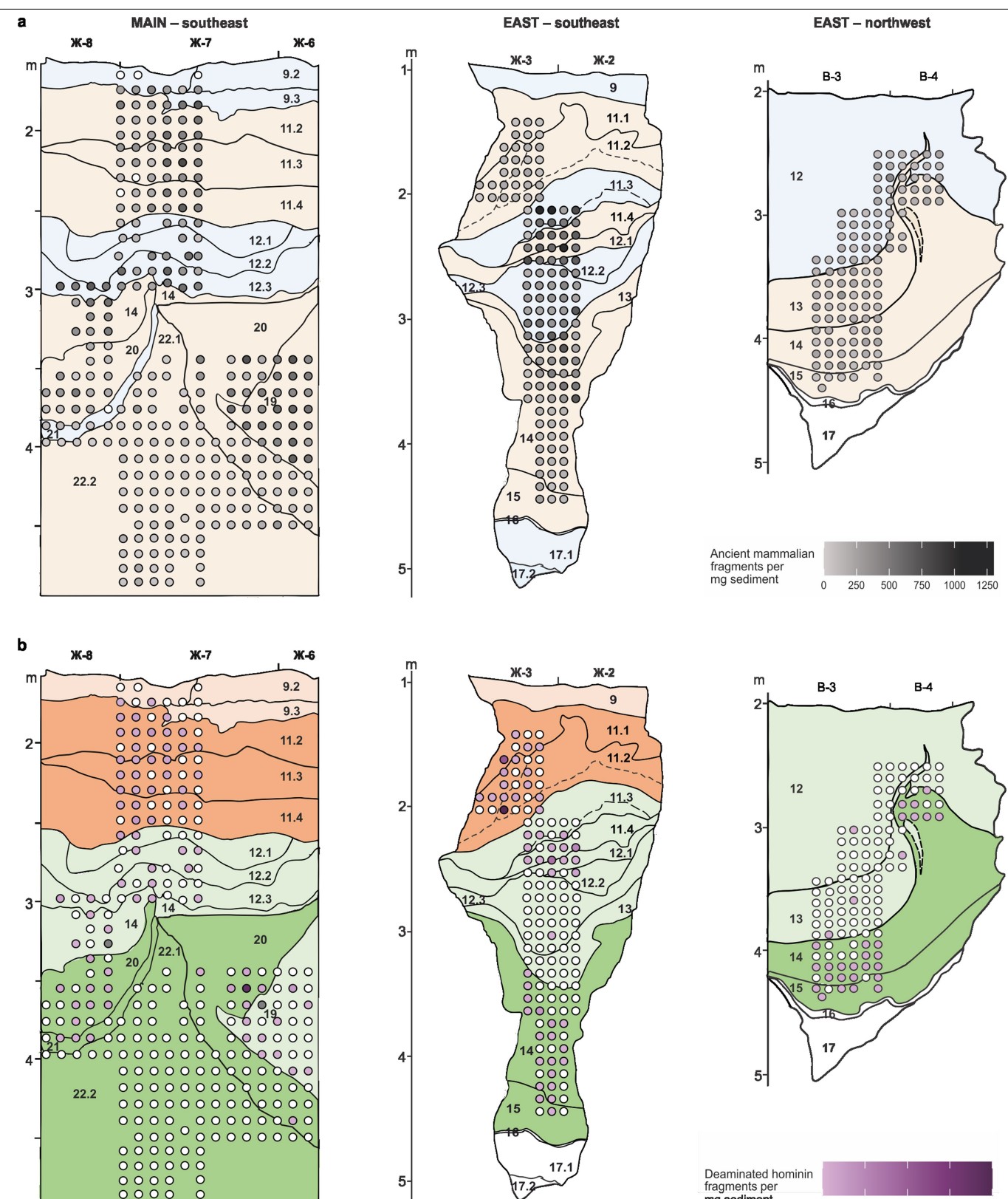

**Extended Data Fig. 5 | Abundance of ancient mammalian and hominin mtDNA in Main and East Chambers. a**, **b**, Number of unique mtDNA fragments assigned to mammalian taxa (**a**) and number of deaminated fragments assigned to hominins (**b**) per milligram of sediment from each library, for samples in Main Chamber (southeast profile) and East Chamber (southeast and northwest profiles). Ancient mammalian fragments encompass all mtDNA fragments assigned to families deemed to contain ancient DNA on the basis of signals of cytosine deamination. Intensity of shading in filled circles reflects relative abundance; samples that yielded no ancient mtDNA are shown as white circles. Background shading in **a** denotes layers that were deposited under relatively cold (blue) or relatively warm (orange) conditions[8], and in **b** denotes layers associated with early Middle Palaeolithic (dark green), middle Middle Palaeolithic (light green), Initial Upper Palaeolithic (dark orange) and Upper Palaeolithic (light orange) archaeological assemblages[8].

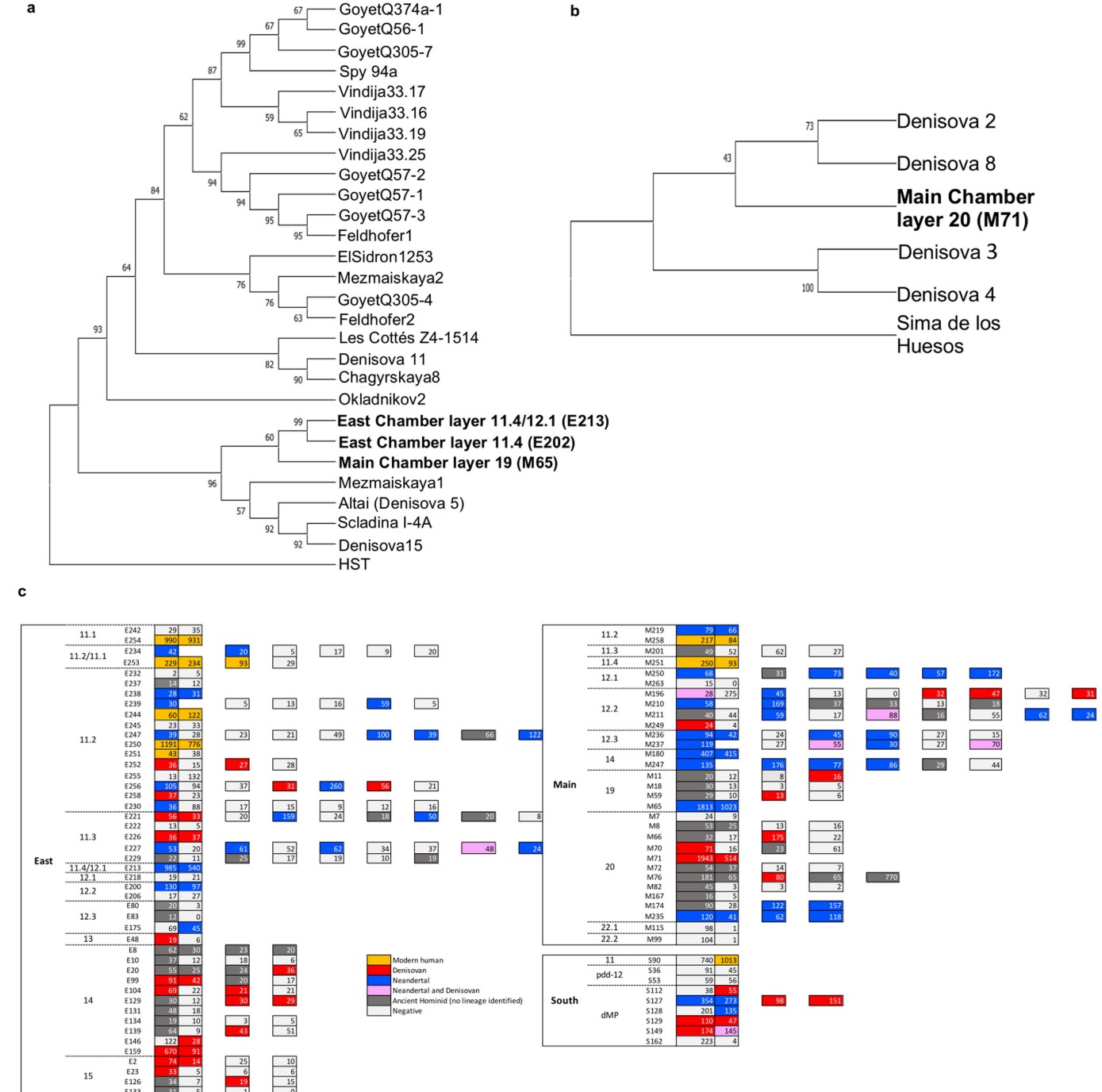

**Extended Data Fig. 6 | Phylogenetic trees for mtDNA sequences assigned to specific hominin lineages, and consistency of hominin mtDNA recovery from individual sediment samples. a**, **b**, Neighbour-joining tree constructed with newly constructed Neanderthal mtDNA consensus sequences (in bold) and published Neanderthal mtDNA genomes (protein-coding genes only) (**a**) and published Denisovan and Sima de los Huesos mtDNA genomes and a newly reconstructed Denisovan mtDNA consensus (protein-coding genes only) (**b**). Taxa were clustered on the basis of pairwise differences, with support for each

node based on a bootstrap test with 500 replicates. Positions with missing data or gaps were removed, resulting in 9,219 and 7,495 positions used for analysis in **a** and **b**, respectively. **c**, Recovery of ancient hominin mtDNA fragments and identification of hominin lineages in cases in which several libraries were prepared from the same sediment sample. Each rectangle represents one library and shows the corresponding number of deaminated fragments. Libraries produced from the same sediment subsample are shown by adjacent rectangles; libraries from different subsamples are kept separate.

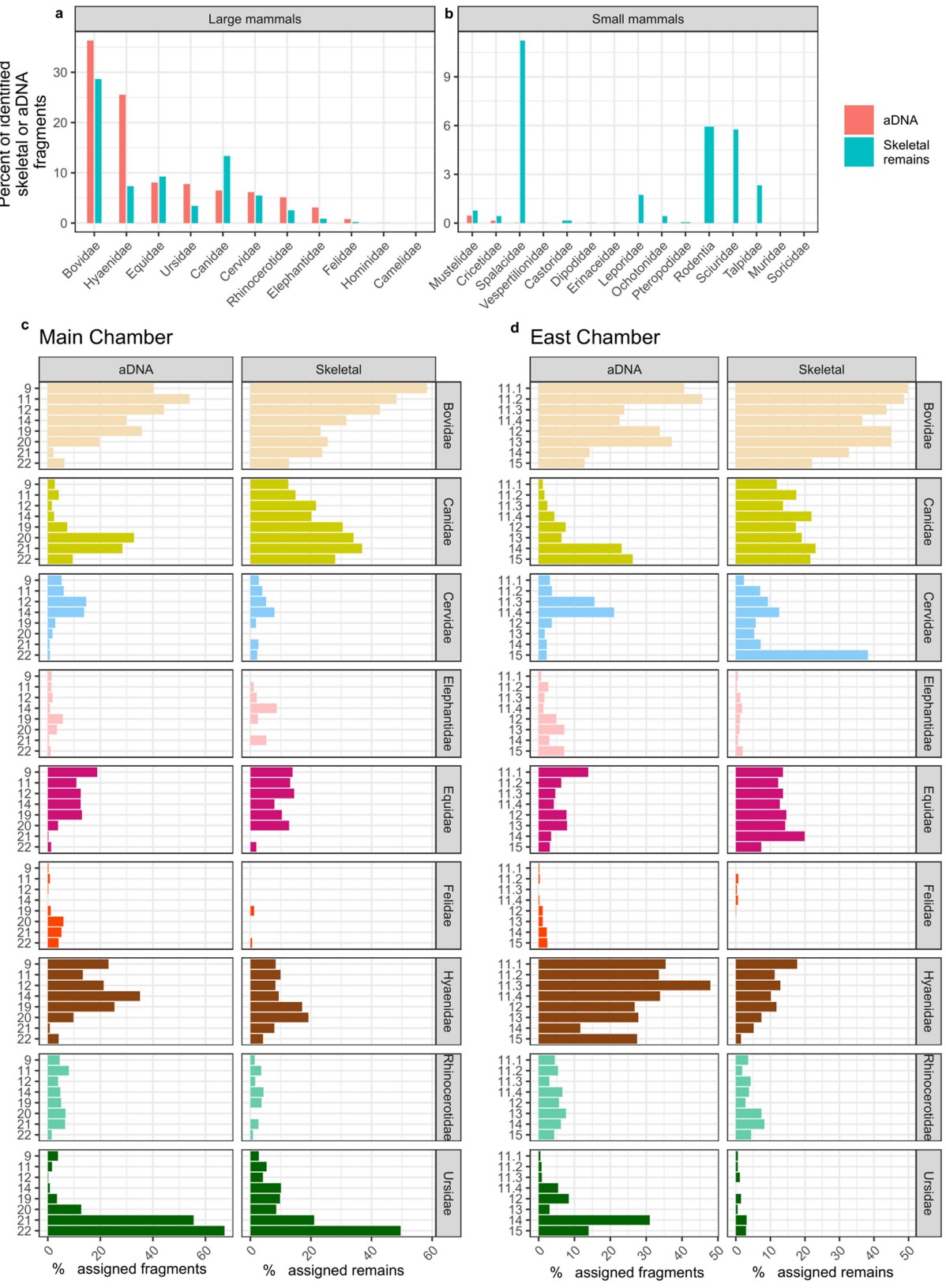

**Extended Data Fig. 7 | Proportions of mammalian mtDNA fragments and skeletal remains in Main and East Chambers. a**, **b**, Proportion (in per cent) of ancient DNA fragments (red) and skeletal remains (blue)[8] assigned to various families of large (**a**) and small (**b**) mammals, combined for all samples in Main and East Chambers and ranked in descending order of mtDNA percentage. **c**, **d**, Proportion (in per cent) of ancient mtDNA fragments and skeletal remains assigned to families of large mammals for individual layers in Main Chamber (**c**) and East Chamber (**d**), arranged in relative stratigraphic order. The mtDNA values were obtained by averaging across the percentages of fragments assigned to each family in all samples from a specified layer.

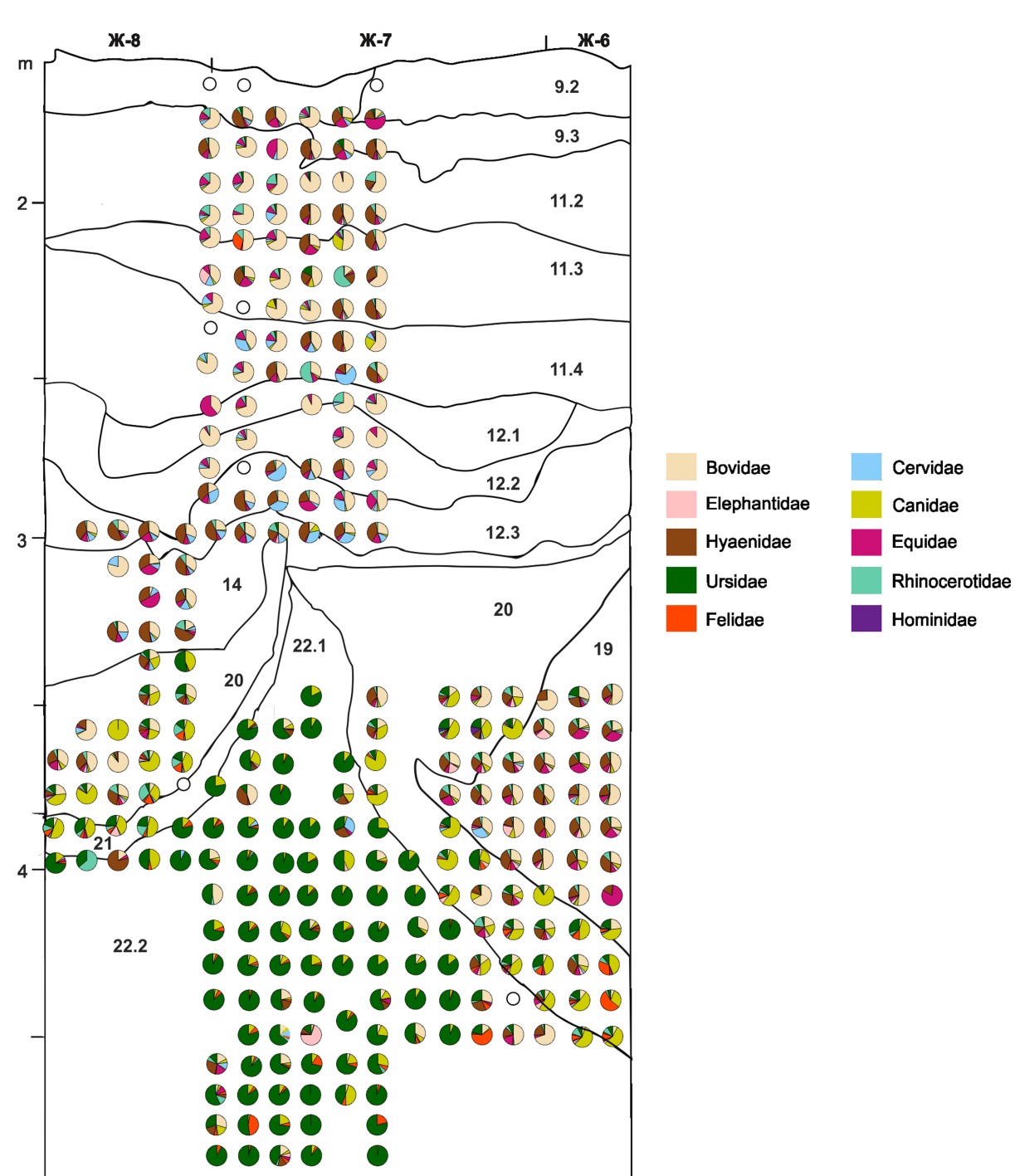

**Extended Data Fig. 8 | Proportions of ancient mtDNA fragments of large mammals in Main Chamber.** Pie charts showing proportions of mtDNA fragments assigned to specific mammalian families for each sample; empty circles denote samples that yielded no ancient mtDNA.

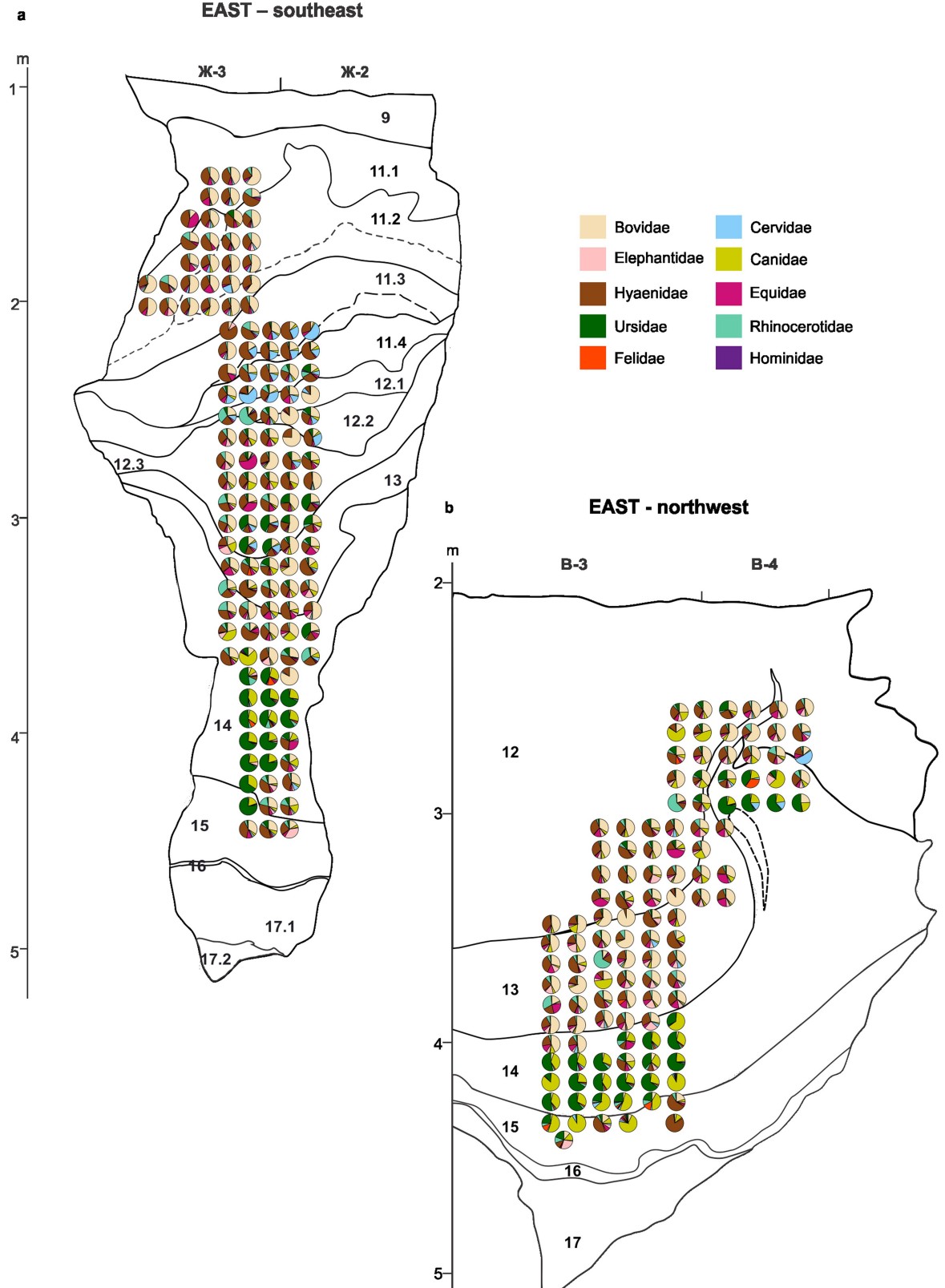

**Extended Data Fig. 9 | Proportions of ancient mtDNA fragments of large mammals in East Chamber. a, b,** Pie charts showing proportions of mtDNA fragments assigned to specific mammalian families for each sample in the southeast (**a**) and northwest (**b**) profiles.

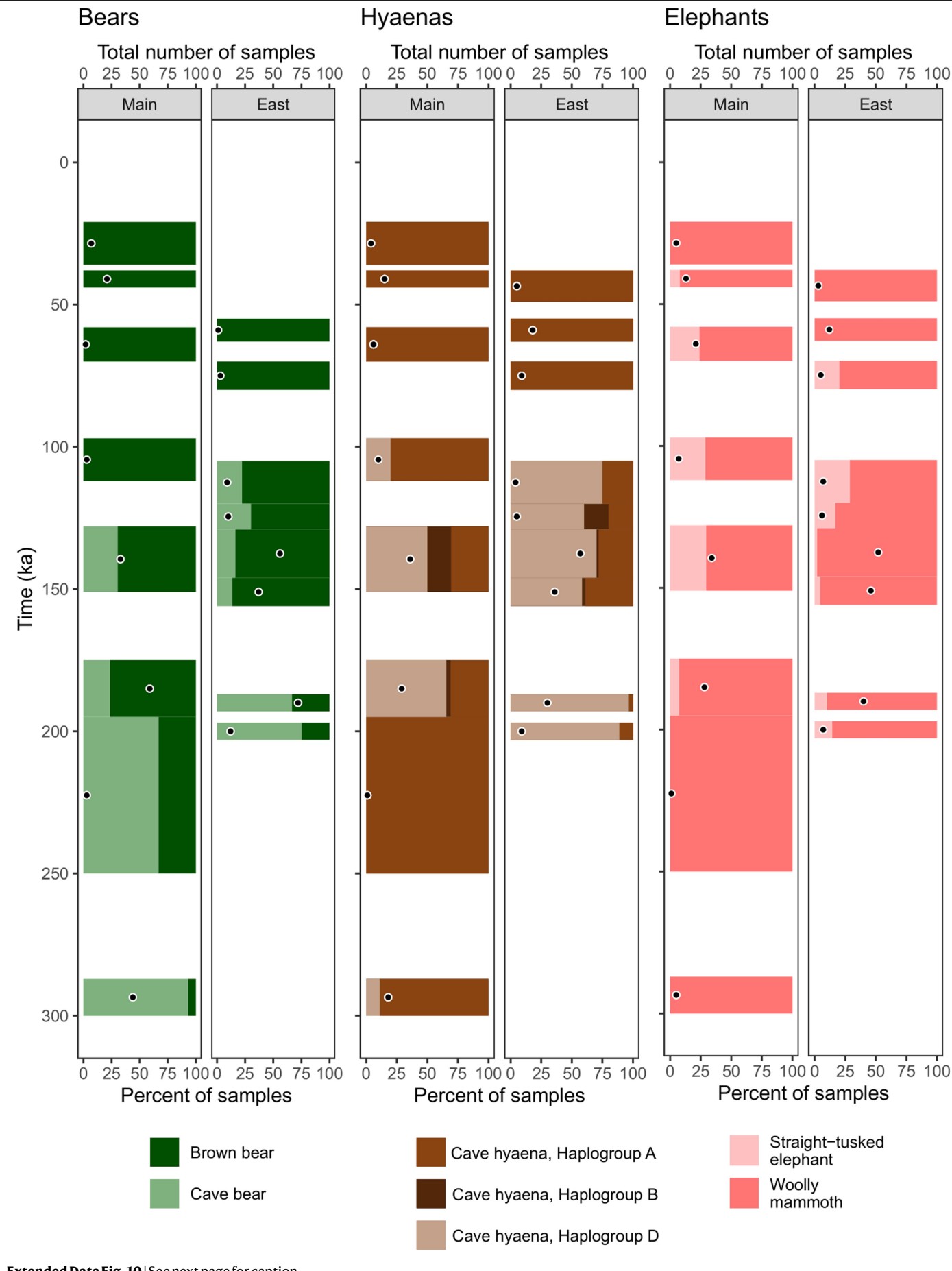

**Extended Data Fig. 10** | See next page for caption.

**Extended Data Fig. 10 | Numbers and proportions of ancient mtDNA fragments assigned to different mtDNA groups of ursids, hyaenids and elephantids in Main and East Chambers.** Black-and-white circles denote the total number of samples assigned to a specific mtDNA clade, plotted as a function of time (mean age for the relevant layer). The corresponding bars are colour-coded on the basis of the proportion (in per cent) of samples assigned to each group, and extend over the time intervals for the respective layers in Main and East Chambers.

# nature research

|  | |
|---|---|

# Reporting Summary

Nature Research wishes to improve the reproducibility of the work that we publish. This form provides structure for consistency and transparency in reporting. For further information on Nature Research policies, see our Editorial Policies and the Editorial Policy Checklist.

## Statistics

For all statistical analyses, confirm that the following items are present in the figure legend, table legend, main text, or Methods section.

| n/a | Confirmed | |
|---|---|---|
| ☐ | ☒ | The exact sample size (*n*) for each experimental group/condition, given as a discrete number and unit of measurement |
| ☐ | ☒ | A statement on whether measurements were taken from distinct samples or whether the same sample was measured repeatedly |
| ☐ | ☒ | The statistical test(s) used AND whether they are one- or two-sided<br>*Only common tests should be described solely by name; describe more complex techniques in the Methods section.* |
| ☐ | ☒ | A description of all covariates tested |
| ☐ | ☒ | A description of any assumptions or corrections, such as tests of normality and adjustment for multiple comparisons |
| ☐ | ☒ | A full description of the statistical parameters including central tendency (e.g. means) or other basic estimates (e.g. regression coefficient) AND variation (e.g. standard deviation) or associated estimates of uncertainty (e.g. confidence intervals) |
| ☐ | ☒ | For null hypothesis testing, the test statistic (e.g. $F$, $t$, $r$) with confidence intervals, effect sizes, degrees of freedom and $P$ value noted<br>*Give P values as exact values whenever suitable.* |
| ☐ | ☒ | For Bayesian analysis, information on the choice of priors and Markov chain Monte Carlo settings |
| ☒ | ☐ | For hierarchical and complex designs, identification of the appropriate level for tests and full reporting of outcomes |
| ☒ | ☐ | Estimates of effect sizes (e.g. Cohen's *d*, Pearson's *r*), indicating how they were calculated |

*Our web collection on statistics for biologists contains articles on many of the points above.*

## Software and code

Policy information about availability of computer code

| Data collection | No software was used for the collection of data. |
|---|---|
| Data analysis | Data was analysed using the leehom package that is available at  https://bioinf.eva.mpg.de/, bam-rmdup package that is available at https://github.com/mpieva/biohazard-tools, MEGAN, BLAST, kallisto and R version 3.5.1. |

For manuscripts utilizing custom algorithms or software that are central to the research but not yet described in published literature, software must be made available to editors and reviewers. We strongly encourage code deposition in a community repository (e.g. GitHub). See the Nature Research guidelines for submitting code & software for further information.

## Data

Policy information about availability of data

All manuscripts must include a data availability statement. This statement should provide the following information, where applicable:

- Accession codes, unique identifiers, or web links for publicly available datasets
- A list of figures that have associated raw data
- A description of any restrictions on data availability

The mitochondrial consensus sequences reported from Main Chamber, layer 19 (M65), layer 20 (M71), East Chamber layer 11.4 (E202) and layer 11.4/12.1 (E213) are available in Dryad digital repository(doi:10.5061/dryad.k3j9kd567) and the raw data for each all mammalian mtDNA and human mtDNA enriched libraries are available in the European Nucleotide Archive under accession number PRJEB44036.

# Field-specific reporting

Please select the one below that is the best fit for your research. If you are not sure, read the appropriate sections before making your selection.

☒ Life sciences       ☐ Behavioural & social sciences       ☐ Ecological, evolutionary & environmental sciences

For a reference copy of the document with all sections, see nature.com/documents/nr-reporting-summary-flat.pdf

# Life sciences study design

All studies must disclose on these points even when the disclosure is negative.

| | |
|---|---|
| Sample size | No sample size was determined in advance. The total number of samples was the result of a grid-like sampling scheme comprising at least three columns (when possible) designed to cover the Pleistocene layers from all the three chambers in Denisova Cave. The resulting number of 728 samples was determined to be sufficient as each Pleistocene layer in each chamber was sampled at least twice. |
| Data exclusions | No samples were excluded from the study. |
| Replication | The experiment of recovering hominin mtDNA from sediment was replicated from 80 samples by generating two libraries from the same sub-sample. The recovery of hominin mtDNA was also tested by taking varying numbers of independent sediment subsamples (8, 6 and 4) from different sediment samples (5, 10, and 49 respectively).The results of these experiments are summarized in Extended Data Figure 6 and described at length in Supplementary Section 4. |
| Randomization | No randomization was performed as this was not relevant for our study. All samples were evaluated for the presence of ancient faunal and hominin mitochondrial DNA and analysis continued for those that contained ancient DNA. |
| Blinding | Blinding was not relevant for data collection as samples were selected based on their location within the stratigraphy. Blinding was also not relevant for downstream analysis as previously established analysis pipelines was used for the processing of the data and results were interpreted based on expectations from and comparisons to previously published ancient and modern mitochondrial genomes. |

# Reporting for specific materials, systems and methods

We require information from authors about some types of materials, experimental systems and methods used in many studies. Here, indicate whether each material, system or method listed is relevant to your study. If you are not sure if a list item applies to your research, read the appropriate section before selecting a response.

## Materials & experimental systems

| n/a | Involved in the study |
|---|---|
| ☒ ☐ | Antibodies |
| ☒ ☐ | Eukaryotic cell lines |
| ☐ ☒ | Palaeontology and archaeology |
| ☒ ☐ | Animals and other organisms |
| ☒ ☐ | Human research participants |
| ☒ ☐ | Clinical data |
| ☒ ☐ | Dual use research of concern |

## Methods

| n/a | Involved in the study |
|---|---|
| ☒ ☐ | ChIP-seq |
| ☒ ☐ | Flow cytometry |
| ☒ ☐ | MRI-based neuroimaging |

## Palaeontology and Archaeology

| | |
|---|---|
| Specimen provenance | Materials were acquired as part of an agreement of scientific cooperation between the Institute of Archaeology and Ethnography, Siberian Branch of the Russian Academy of Sciences and the Max Planck Institute for Evolutionary Anthropology for projects in the field of palaeogenetics in North Asia, signed on December 25, 2018 and valid for a duration of five years.  The Institute of Archaeology and Ethnography, Siberian Branch of the Russian Academy of Science oversees the excavation of Denisova Cave and obtained all permits necessary for conducting archaeological fieldwork and research associated with this project from the Ministry of Culture of the Russian Federation. |
| Specimen deposition | All sediment samples remain stored at the Max Planck Institute for Evolutionary Anthropology in Leipzig, Germany |
| Dating methods | No new dates are provided in this study. All ages referred to in this paper are from Jacobs et al. (Nature 565, 594–599, 2019) and Douka et al. (Nature 565, 640–644, 2019). |

☐ Tick this box to confirm that the raw and calibrated dates are available in the paper or in Supplementary Information.

| | |
|---|---|
| Ethics oversight | All necessary permits for excavations at Denisova Cave were obtained by the Institute of Archaeology and Ethnography, Siberian Branch of the Russian Academy of Science from the Ministry of Culture of the Russian Federation. |

Note that full information on the approval of the study protocol must also be provided in the manuscript.

