## [Peer Review File · Nature]

Manuscript Title: Pleistocene sediment DNA reveals hominin and faunal turnovers at Denisova Cave

Editorial Notes:

Reviewer Comments & Author Rebuttals

Reviewer Reports on the Initial Version:

Referee #1 (Remarks to the Author):

Review of "DNA from sediments reveals hominin and faunal turnovers at Denisova Cave" by Zavala et al.

In this paper, the authors report a systematic analysis of 728 sediment DNA samples from Denisova Cave, collected in a grid pattern across excavation profiles spanning more than 200,000 years from the three chambers: the South Chamber, the East Chamber, and the Main Chamber. Using state-of-the-art ancient DNA methods that maximize the extraction of DNA from marginal samples, along with in-solution enrichment for mitochondrial DNA sequences, they discover mammalian mitochondrial DNA in 685 sediments and hominin mitochondrial DNA in 175 sediments. A breakthrough study of sedimentary DNA by this group was published several years ago (Slon et al. Science 2017), and included analysis of 52 sediment samples from Denisova Cave. The present study is a major advance yet again, realizing the promise of archaeogenetic analysis of sediments to provide a high resolution characterization of the faunal occupation history of a site. The most important findings of this study relate to the chronicle they provide of the alternate occupations of the site by Denisovans and Neanderthals and eventually modern humans. But the findings about faunal occupation history are also extremely important and enhance the findings about the hominins.

MAJOR ADVANCES AND FINDINGS RELATIVE TO PREVIOUS WORK

(a) The study represents a more than 10-fold increase in the number of sediments analyzed from Denisova Cave. Facilitated by the much larger number of samples, the study can be systematic with regard to sedimentary layers and coverage of the different archaeological profiles, making it possible to ask and answer questions that were not possible to address in the earlier work.

(b) The study reveals population turnovers over time in multiple fauna which shows the extraordinary power of the study of sediment DNA analysis to provide a chronicle of large mammal changes in population structure over time in association with changes in climate

Hominins:

250-150 kya Denisovans of the Denisova2/8 mtDNA type

170-100 kya Neanderthals carrying the Scladina clade haplogroup

130-100 kya Only Neanderthals detected suggesting Denisovans disappeared

80-45 kya Denisovans reappear

80-45 kya Neanderthals carrying the Denisova 11 clade haplogroup

<45 kya Modern humans appear

Ursids:

>187 kya cave bears

<112 kya brown bears

Hyenas:

>200 kya clade A

200-120/80 kya clade D

<80 kya clade A

Changes in proportions of mtDNA from many different taxa (including increases in proportions of bovids and hyaenids and decreases in proportions of ursids and canids) around ~190 kya corresponding to the MIS7/MIS6 climate transition from warm to cold

Changes in proportions of mtDNA from many different taxa (including disappearance of Denisovans and decline in bovid, ursid, felid, and canid proportions) around ~130kya corresponding to the MIS6/MIS5 climate transition from cold to warm (the beginning of the last interglacial).

Changes in mtDNA distributions during the MIS5/4 distribution from warm to cold, including the appearance of late Neanderthal haplogroups and reappearance of hyaenid clade A.

MINOR SUGGESTIONS

(1) It would be nice to show in Figure 1d the phylogenetic positions of all four of the assembled mtDNA sequences from the new sediment samples.

(2) The authors write:

“Among all 223 samples from eMP layers in Main 148 and East Chambers, 50 contained evidence for Denisovan mtDNA and only three, from layer 20 in Main Chamber, for Neanderthal mtDNA. Two of these (M174 and M235) contain typical Neanderthal mtDNA and are from areas where small-scale mixing with overlying sediments may have occurred, whereas the third (M76) is from the middle of the layer and carries the previously unknown Neanderthal mtDNA lineage. These results point to Denisovans as the first and principal makers of the eMP assemblages, which are older than 170 ± 19 ka, but also provide evidence for first Neanderthal occupation of Denisova Cave towards the end of the eMP, thereby raising the possibility that Neanderthals may have contributed to the production of these assemblages in their later stages.”

The fact that the one eMP sequence that clades with Neanderthals is basal to all Neanderthal sequences studied to date provides evidence of its authenticity, and suggests an ultimate Neanderthal origin. While I agree that this does provide evidence of eMP Neanderthal presence in the region, one cannot determine group identity based on a single locus (mtDNA), and I think the authors might want to also mention the possibility that this could be a Neanderthal sequence introgressed into Denisovans from a neighboring population (not living in the Denisova Cave region itself). The Denisova 11 individual had a Neanderthal mother and Denisovan father, so if an early Middle Paleolithic individual like this occurred somewhere and had offspring with Denisovans, it could contribute to a Denisovans carrying Neanderthal mtDNAs.

(3) Supplementary Section 3 is an impressive analysis of how different covariates affect sediment preservation. However, the only analyses that are presented are ones that examine one independent variable at a time (stratigraphic depth, sediment pH, post-depositional phosphatization, clast size, cave chamber, and sediment color). Sediment depth clearly has a profound effect, so it would be good to control for this effect in analysis of the other features. For example, in Supplementary Figure 11, what is the effect of sediment color *after* controlling for the effect of sediment depth? It seems possible to me that sediment colors are non-randomly distributed according to sediment depth. I think a multiple regression analysis could be informative, as it would show the effect of each independent variable after controlling for the others.

(4) For the branch shortening estimate of the age of sample M65 in Supplementary Section 7, I am a little worried that the process of building a consensus sequence for the mitochondrial DNA from this sample might result in artifactual branch shortening. In Supplementary Table 7 there is evidence that there are 2 or more individuals contributing to sample M65. At sites where these two individuals have sequences that differ from each other, I assume that a consensus is less likely to be called, and this will result in the elimination of terminal mutations specific to the sample in the consensus sequence, causing the sample to artifactually seem too old. Is it possible that this is going on? If it's

not an issue, the authors might wish to add a sentence explaining why this is the case.

(5) In the first line of Supplementary Section 8, there is a reference to "Supplementary Section X". What is "X". Also, there is a reference to the program kallisto (reference 61) which doesn't exist in the reference list (the reference list jumps from 60 to 62).

Referee #2 (Remarks to the Author):

This is a landmark paper which was a pleasure to read and comment on.

The authors report the results of a large-scale, systematic and extensive analyses of sediment for the recovery of ancient DNA, from Denisova Cave, a Pleistocene age site in the Siberian Altai. The site has yielded numerous remains of Neanderthal and Denisovans and excellent biomolecular preservation makes it an ideal canvas for the application of this type of work.

While similar analyses have been applied previously at the site, this time the automation of the workflow allowed a larger number of individual samples be analysed. Starting with the workload, their collection, recording and analyses of 728 sediment samples for DNA is enormous and the authors are to be praised for their consistent and meticulous work. Of these, 685 samples (94%) yielded faunal DNA and 175 (24%) yielded hominin DNA (Denisovan, Neanderthal and modern human) too. Very interestingly the later % is very similar to what was previously reported by Slon et al. (there 12 of 52 samples yielded hominin DNA).

Four samples yielded enough mtDNA sequences for at least 3-fold coverage of the mt genome and resulted in the reconstruction of phylogenetic trees and determination of the relationship between the sediment-deriving hominin mtDNA to the mtDNAs of archaic human fossils. The mtDNA sequences of 3 samples (East Chamber, layer 11.4 and 11.4/12.1, and Main Chamber, layer 19) align with Neanderthal mtDNA and the fourth sequence (Main Chamber, layer 20) is of Denisovan type. One of the Neanderthal mtDNA sequence comes from a previously unknown Neanderthal branch that diverged ~250-230 ka ago. This finding offers significant insights in the early development and evolution of Neanderthals and Denisovans and finding out more about this particularly Neanderthal branch I am sure will form the basis of further new investigations.

Very importantly and with major archaeological implications is the recovery of ancient modern human mtDNA in the Upper Palaeolithic layers of the site. This is for the first time since no modern human fossils have been identified at the site so far. Pleistocene-age modern human remains are very sparse in the Altai, and in northern Eurasia as a whole, so the detection of modern human aDNA is an exciting development that will revolutionise the way we excavate and analyse our sites.

I cannot comment on the validity of the aDNA statistics and methods, however the conclusions are based on the findings and appear robust. The references are appropriate and the abstract, introduction and conclusions are clear and to the point.

My slight criticism to the text is that the first 2 paragraphs of the discussion repeat what has already been said previously in the text. I am not sure how necessary it is to do so.

Finally, I find the presentation of the data, in particular the figures rather complicated, each been composed by several plates and, generally, difficult in the eye. I think the authors need to step back and decide their priorities on what exactly they want to present as their main findings and place the rest in the SI or remove them altogether.

Figure 1: There's too much information and many datapoints and demarkations to take in (black numbers, red numbers; red, blue circled numbers for different fossils; dashed lines; background shading; filled circles with red, blue, yellow, grey, empty circles; crosses, stars, diamonds, triangles). I suggest simplifying it.

I don't particularly think providing modelled start and end ages on the left side of the columns is

necessary, they clutter the figure even more. Maybe best include the total span of each archaeological phase (eMP, mMP, IUP). Plates a, b and c could remain in same figure, with (d) forming a new Fig. 3, possibly alongside plates (a) and (b) from Extended data Fig. 6?

I'm afraid Extended data figures are way too complex.

Extended Data Fig. 1 is a repeat of what has already been published by Douka et al, and Jacobs et al. 2020. Why include it here when nothing in it is new? I suggest removing it or at best move to the SI. Equally the mammalian composition ext data figures are too many. A simplified version of these figures might be:

Start with what is now Extended data Fig. 2 and turn it in Extended data Fig. 1, followed by Extended data Fig. 5 (which sort of synthesizes the findings of the paper). What appears as plate (c) in Extended data Fig. 6 should either be combined with, or follow, Extended data Fig. 5. Extended data Fig. 4, 7 and 10 appear to me SI material.

A few more specific comments:

- Lines 42-44: I suggest that the authors simplify this paragraph. Maybe best state the earliest ages for the occupation in each Chamber, and then state the span for the eMP, mMP and IUP irrespective to Chamber.

- Line 44: Initial occupation starts in the East Chamber is in layer 15 which dates to ~200ka. Please check the original Jacobs et al. 2020 publication and correct accordingly.

- Line 80: what's the median? If you include it in line 99 you should include it here too.

- Line 165: I think it should be Last Interglacial (with capitals)

Referee #3 (Remarks to the Author):

Zavala and colleagues present a fine-resolution reconstruction of mammalian and hominin occurrences and turnovers at Denisova Cave, from the Middle to Late Pleistocene, using sedimentary ancient DNA (sedaDNA) from >700 samples.

This is an excellent, enjoyable, and well-written paper. The careful experimental design, which includes systematic sampling and spatial replication within and between cave chambers, together with new methods for robustly identifying hominin mitochondrial lineages has allowed the authors to thoroughly explore the mammalian and hominin history of Denisova cave. The authors have additionally used the data set to examine DNA preservation against different sediment characteristics and time at this single locality. The distinct stratigraphic partitioning of the recovered hominin and mammalian DNA suggests that leaching and reworking - particular concerns for cave sediments - are not a major influence on the main results.

Of novelty, the authors identify a new neanderthal mitochondrial lineage and have developed a type of sedaDNA-based biostratigraphy. The findings of this latter development are supported by the skeletal record.

The methodology, data, analyses, and interpretations appear to be mostly sound. My concerns about the paper in its present form are as follows:

The entire manuscript, including the final paragraph, is focused on Denisova Cave. However, the approaches developed and showcased here have much wider applicability to other sites, taxa, and substrates. I suggest that the authors add a new concluding paragraph and sentence to the abstract to highlight the broad utility of their findings beyond this single locality.

Consider inserting 'Pleistocene' into the title, as there is currently no temporal dimension mentioned.

L77-94: The authors test DNA preservation (as measured by amount, fragment size, and extent of 5' deamination) against time and multiple sediment characteristics. For two of these measures (amount and fragment size), the authors use taxonomically-combined data. For the deamination analyses, however, data are partitioned based on hominin and four mammalian families. What is the rationale for taxonomically splitting the deamination analyses and not, for example, the fragment size analyses? Given that this is sedaDNA derived from taxa with similar tissue and cellular structures, would it not be safe to assume that deamination and fragment size profiles from the sedaDNA are independent of taxonomic origin? In fact, the authors could test whether the four mammalian families have significantly different deamination profiles within samples. Assuming the results of this test are non-significant (as hinted at by e.g. Supplementary Figure 6), then I suggest that the deamination comparison analyses are performed on the taxonomically-combined data. This would increase statistical power and reduce the number of comparisons presented in Extended Data Figures 3 and 4, and SI Section 3.

L85-86,561,578: what does a 'significant increase' mean here? How was this tested or was a threshold used?

L155-157: I thank the authors for formally correcting this error in their pilot study. One other mismatch in the literature exists; the layer from which Denisova-9 was recovered (Jacobs et al. 2019, Fig. 3: layer 12.2; Douka et al. 2019, Fig. 3: layer 12.3). The authors follow Jacobs et al. in Figure 1 - is this correct?

L167-175: Modern human mitochondrial DNA was recovered from samples taken from the IUP and UP layers. For three of these samples, a single modern human contributor was determined (Supplementary Table 7). What are the mitochondrial haplogroups for these three samples? Assuming these assignments make phylogeographic and temporal sense, then this would provide additional supporting evidence for result authenticity.

L167-175: There seems to be only a single unreplicated sample with modern human mitochondrial DNA from the UP layer (Figure 1b). Do the authors consider this finding robust, given the potential for reworking from IUP layers? If not, I suggest the authors state this limitation.

L173-175: Denisovan, neandertal, and modern human DNA are all found in layer 11.2 from the East Chamber, suggesting that all three groups were present in the area during the IUP. However, could this be an artefact of sediment reworking/mixing in this layer? Is there other evidence (chronological, stratigraphic, etc) to suggest that this is not the case?

L191-232: I think the authors can be bolder with their mammalian data, as they have effectively devised a sedaDNA-based biostratigraphy for Denisova Cave. For example, the author's could hypothesize that layer 22 from the South chamber is of early Middle Palaeolithic age based on the predominance of ursid (cave bear) DNA (Supplementary Figure 27; Supplementary Data File 1), which is characteristic of the eMP in the other two chambers (Extended Data Figures 8 and 9).

L208-220: Three mammalian families (ursids, elephantids, and hyaenids) were selected based on 'availability of complete mitochondrial genome sequences that cover the genetic diversity of extant and extinct species within a family' (pg45 of SI). However, this is also comparatively true for bovids and equids. Any reason to exclude these two families?

From the three selected families, the authors exclude identified taxa that are unlikely to have been in the region (e.g. polar bear, Columbian mammoth). However, they include 'African and straight-tusked elephant' (Extended Data Figure 10; pg46 of SI). As African elephants are not known outside of Africa (including in the fossil record), the authors would be on safe ground to refer to this mtDNA group simply as 'straight-tusked elephant'.

L224-225: for ecological context, state that this transition from an interglacial (MIS 7) to glacial (MIS

6) lead to climatic deterioration.

L227: the last interglacial is MIS 5e only (123-118 ka). Perhaps rephrase to 'during the climatic amelioration from MIS 6 to MIS 5' or similar.

L241-242: How confident are the authors that their sedaDNA data are accurately reflecting relative abundance of large mammals? There seems to be a possible correlation with relative abundance of skeletal remains (Extended Data Figure 7), although the strength of this is not formally tested. Note that taphonomic and ecological factors, such as if the cave was periodically used as a predator den, could bias relative abundance estimates.

Figure 1: In panel a, should Denisova-9 be from layer 12.2 (following Jacobs et al. 2019) or 12.3 (following Douka et al. 2019)? In panel a, samples 251 and 244 should be modern human only (based on Extended Data Figure 6).

Figure 2: In panel a, % of biogenic silica is presented as an environmental proxy, but with no explanation in the figure legend. I assume the authors are using this as a proxy for local temperature? If so, is there good reason to use this record instead of the Greenland oxygen isotope record?

There seems to be conflicting warm or cold conditions inferred from the same time intervals between the Main and East chambers. Perhaps add a sentence describing the sources of uncertainty here.

Extended Data Figure 1: perhaps add a panel with the position and assignment of hominin sedaDNA from the Slon et al. pilot study.

Extended Data Figure 2: what do the alphanumeric symbols at the left and top of panel a represent?

Extended Data Figure 3: State here that Layer 11 is potentially IUP. The phosphate deformation deposits are stratigraphically split into pdd-9 and pdd-12, but this division is not highlighted in panel a. Please also make the same correction to Supplementary Figure 3.

Extended Data Figure 4: L379-381: are the test results for the average fragment size and number of fragments mixed up? The highest p-value is reported for average fragment size, but this shows a clearer decrease than number of fragments. Check also SI Section 3.1 (pg 9).

Extended Data Figure 7: the panel letters are missing from the figure. In c and d, the x-axes are correct for aDNA, but should be 'percent of assigned remains' for skeletal. Some of the family names are truncated in c and d (note that this is also an issue for Supplementary Figures 21 and 22).

Extended Data Figure 10: I suggest renaming the hyaenids to be consistent with the other two families. For example 'Cave Hyaena, Hg B' instead of 'Haplogroup B'.

Minor:

L32-33: give the approximate age of the onset of the Initial Upper Palaeolithic

L38: give the age range in absolute time here as well, for readers not familiar with the ages of the Pleistocene and Holocene.

L60: archaic humans or archaic hominins?

L68,81: check order of the Extended Data Figures

L197: taxonomic family names should not be italicized

L463: how far was the exposed profile cleaned back?

L514: how many copies of the control oligo were added to each sample?

L536-537: what read length was used? paired- or single-end?

Supplementary Information:

pg5, S2.1: layer 11.5 is also not exposed
pg7, S2.3: typo: 'ppd'
pg9, S3.1: check order of statistical tests is correct (see above)
pg30, S7.2: Note that, in BEAST, groups can be selected to estimate tMRCA without enforcing monophyly.
pg33, S8: typo: 'Supplementary Section X'
pg33, S8: typo: 'Using the full set of mtDNA genomes is used'
pg33, S8: give reference for '[Vernot et al, in press]'
pg39, S9: consider re-writing the second paragraph to improve readability, perhaps just by giving the range of midpoint branch supports. The CI info could then be given in Supplementary Figure 24.
pg39, S9: typo: 'indicating the presence different'
pg46, S11: typo: '143'
pg46, S11: typo: 'beat'

Supplementary Figure 1: clarify in the legend that two separate samples were taken for two locations.

Supplementary Figure 3: more information is needed here on layers 11 and 22. Presumably 'deformed MP' means 'deformed Middle Palaeolithic'? And presumably layer 22 is older? How much younger is layer 11? And are the layers that include the 'phosphate deformation deposits' older or younger than layer 11? Please clearly indicate the Holocene deposits.

Supplementary Figure 6: it is not necessary to also display the black outlier dots, given that measurement data are already overlain.

Supplementary Figure 16: the legend states: 'The chimpanzee branch used to root the tree is not shown.', but this is shown.

Supplementary Figure 20: please improve the resolution of this figure.

Supplementary Figure 27: add a colour-to-mammalian-family lookup key to the figure.

Supplementary Data File 1, sheet 'General Sample Summary': please add the European Nucleotide Archive sample accessions here.

Supplementary Data File 2: check cell H103.

Supplementary Data File 3, sheet 'Mammalian mtDNA clade refs':
rename Arctotherium clade from 'Arctic bear' to 'shortface bear'. Arctotherium is the South American short-faced bear (Arctos is the Greek word for bear).
why are the elephantid clades in latin, whereas common names are used for the ursids and hyaenids?

Note that Supplementary Data Files 2 and 3 were uploaded in the wrong order.

Author Rebuttals to Initial Comments:

Referee #1 (Remarks to the Author):

Review of "DNA from sediments reveals hominin and faunal turnovers at Denisova Cave" by Zavala et al.

In this paper, the authors report a systematic analysis of 728 sediment DNA samples from Denisova Cave, collected in a grid pattern across excavation profiles spanning more than 200,000 years from the three chambers: the South Chamber, the East Chamber, and the Main Chamber. Using state-of-the-art ancient DNA methods that maximize the extraction of DNA from marginal samples, along with in-solution enrichment for mitochondrial DNA sequences, they discover mammalian mitochondrial DNA in 685

sediments and hominin mitochondrial DNA in 175 sediments. A breakthrough study of sedimentary DNA by this group was published several years ago (Slon et al. Science 2017), and included analysis of 52 sediment samples from Denisova Cave. The present study is a major advance yet again, realizing the promise of archaeogenetic analysis of sediments to provide a high resolution characterization of the faunal occupation history of a site. The most important findings of this study relate to the chronicle they provide of the alternate occupations of the site by Denisovans and Neanderthals and eventually modern humans. But the findings about faunal occupation history are also extremely important and enhance the findings about the hominins.

MAJOR ADVANCES AND FINDINGS RELATIVE TO PREVIOUS WORK

(a) The study represents a more than 10-fold increase in the number of sediments analyzed from Denisova Cave. Facilitated by the much larger number of samples, the study can be systematic with regard to sedimentary layers and coverage of the different archaeological profiles, making it possible to ask and answer questions that were not possible to address in the earlier work.

(b) The study reveals population turnovers over time in multiple fauna which shows the extraordinary power of the study of sediment DNA analysis to provide a chronicle of large mammal changes in population structure over time in association with changes in climate

Hominins:

250-150 kya Denisovans of the Denisova2/8 mtDNA type

170-100 kya Neanderthals carrying the Scladina clade haplogroup

130-100 kya Only Neanderthals detected suggesting Denisovans disappeared

80-45 kya Denisovans reappear

80-45 kya Neanderthals carrying the Denisova 11 clade haplogroup

<45 kya Modern humans appear

Ursids:

>187 kya cave bears

<112 kya brown bears

Hyenas:

>200 kya clade A

200-120/80 kya clade D

<80 kya clade A

Changes in proportions of mtDNA from many different taxa (including increases in proportions of bovids and hyaenids and decreases in proportions of ursids and canids) around ~190 kya corresponding to the MIS7/MIS6 climate transition from warm to cold

Changes in proportions of mtDNA from many different taxa (including disappearance of Denisovans and decline in bovid, ursid, felid, and canid proportions) around ~130kya corresponding to the MIS6/MIS5 climate transition from cold to warm (the beginning of the last interglacial).

Changes in mtDNA distributions during the MIS5/4 distribution from warm to cold, including the appearance of late Neanderthal haplogroups and reappearance of hyaenid clade A.

MINOR SUGGESTIONS

(1) It would be nice to show in Figure 1d the phylogenetic positions of all four of the assembled mtDNA sequences from the new sediment samples.

We appreciate this suggestion to consolidate all the genetic information into a single figure. However, the tree shown in Figure 1d includes only mtDNA genomes used as references for identifying a subset of specific Neanderthal and Denisovan mitochondrial lineages that are represented by symbols in panels a–c (with the exception of the Neanderthal sample from layer 20), not the full phylogeny. Complete trees that include all Neanderthal and Denisovan mtDNA sequences currently available are presented in Extended Data Figure 6, panels a and b. We have added information to the legend of Figure 1 to clarify this.

(2) The authors write:

“Among all 223 samples from eMP layers in Main 148 and East Chambers, 50 contained evidence for Denisovan mtDNA and only three, from layer 20 in Main Chamber, for Neanderthal mtDNA. Two of these (M174 and M235) contain typical Neanderthal mtDNA and are from areas where small-scale mixing with overlying sediments may have occurred, whereas the third (M76) is from the middle of the layer and carries the previously unknown Neanderthal mtDNA lineage. These results point to Denisovans as the first and principal makers of the eMP assemblages, which are older than 170 ± 19 ka, but also provide evidence for first Neanderthal occupation of Denisova Cave towards the end of the eMP, thereby raising the possibility that Neanderthals may have contributed to the production of these assemblages in their later stages.”

The fact that the one eMP sequence that clades with Neanderthals is basal to all Neanderthal sequences studied to date provides evidence of its authenticity, and suggests an ultimate Neanderthal origin. While I agree that this does provide evidence of eMP Neanderthal presence in the region, one cannot determine group identity based on a single locus (mtDNA), and I think the authors might want to also mention the possibility that this could be a Neanderthal sequence introgressed into Denisovans from a neighboring population (not living in the Denisova Cave region itself). The Denisova 11 individual had a Neanderthal mother and Denisovan father, so if an early Middle Paleolithic individual like this occurred somewhere and had offspring with Denisovans, it could contribute to a Denisovans carrying Neanderthal mtDNAs.

This is a good point and we agree that it is important to discuss the possibility of gene flow. We have rephrased the text accordingly (lines 157–158).

(3) Supplementary Section 3 is an impressive analysis of how different covariates affect sediment preservation. However, the only analyses that are presented are ones that examine one independent variable at a time (stratigraphic depth, sediment pH, post-depositional phosphatization, clast size, cave chamber, and sediment color). Sediment depth clearly has a profound effect, so it would be good to control for this effect in analysis of the other features. For example, in Supplementary Figure 11, what is the effect of sediment color *after* controlling for the effect of sediment depth? It seems possible to me that sediment colors are non-randomly distributed according to sediment depth. I think a multiple regression analysis could be informative, as it would show the effect of each independent variable after controlling for the others.

Thank you for reading this section so closely and we are glad that you found it interesting. We have added multiple regression and anova tests for each comparison that was identified as significant, to control for the impact of time (as represented by the layer from which each sample was collected). In all cases, the significance of the correlation decreased, but did not disappear. We also added a sentence in Supplementary Section 3.5 (DNA preservation and sediment colour) pointing out that the blackened, yellow and rusty ochre sediments (each stated as correlating with fewer recovered ancient DNA fragments and shorter fragment sizes) are all from the lowest layers of the cave, so we can't separate them from the impact of time.

(4) For the branch shortening estimate of the age of sample M65 in Supplementary Section 7, I am a little worried that the process of building a consensus sequence for the mitochondrial DNA from this sample might result in artifactual branch shortening. In Supplementary Table 7 there is evidence that there are 2 or more individuals contributing to sample M65. At sites where these two individuals have sequences that differ from each other, I assume that a consensus is less likely to be called, and this will result in the elimination of terminal mutations specific to the sample in the consensus sequence, causing the sample to artifactually seem too old. Is it possible that this is going on? If it's not an issue, the authors might wish to add a sentence explaining why this is the case.

This is a good question and an issue that also concerned us. We examined the positions that failed consensus calling and found that, for the protein-coding region used for branch shortening analysis, only one position failed due to low consensus support (Supplementary Table 8). For this reason, combined with the determination of one primary mitochondrial haplotype (Supplementary Table 7: Proportion of major haplotype = 0.915), we do not think that this is an issue of concern. We have added an explanation to the end of the concluding paragraph in Supplementary Section 6.

(5) In the first line of Supplementary Section 8, there is a reference to “Supplementary Section X”. What is “X”. Also, there is a reference to the program kallisto (reference 61) which doesn’t exist in the reference list (the reference list jumps from 60 to 62).

Thank you for pointing these out. Both have been fixed.

Referee #2 (Remarks to the Author):

This is a landmark paper which was a pleasure to read and comment on.

The authors report the results of a large-scale, systematic and extensive analyses of sediment for the recovery of ancient DNA, from Denisova Cave, a Pleistocene age site in the Siberian Altai. The site has yielded numerous remains of Neanderthal and Denisovans and excellent biomolecular preservation makes it an ideal canvas for the application of this type of work.

While similar analyses have been applied previously at the site, this time the automation of the workflow allowed a larger number of individual samples be analysed. Starting with the workload, their collection, recording and analyses of 728 sediment samples for DNA is enormous and the authors are to be praised for their consistent and meticulous work. Of these, 685 samples (94%) yielded faunal DNA and 175 (24%) yielded hominin DNA (Denisovan, Neanderthal and modern human) too. Very interestingly the later % is very similar to what was previously reported by Slon et al. (there 12 of 52 samples yielded hominin DNA). Four samples yielded enough mtDNA sequences for at least 3-fold coverage of the mt genome and resulted in the reconstruction of phylogenetic trees and determination of the relationship between the sediment-deriving hominin mtDNA to the mtDNAs of archaic human fossils. The mtDNA sequences of 3 samples (East Chamber, layer 11.4 and 11.4/12.1, and Main Chamber, layer 19) align with Neanderthal mtDNA and the fourth sequence (Main Chamber, layer 20) is of Denisovan type. One of the Neanderthal mtDNA sequence comes from a previously unknown Neanderthal branch that diverged ~250-230 ka ago. This finding offers significant insights in the early development and evolution of Neanderthals and Denisovans and finding out more about this particularly Neanderthal branch I am sure will form the basis of further new investigations.

Very importantly and with major archaeological implications is the recovery of ancient modern human mtDNA in the Upper Palaeolithic layers of the site. This is for the first time since no modern human fossils have been identified at the site so far. Pleistocene-age modern human remains are very sparse in the Altai, and in northern Eurasia as a whole, so the detection of modern human aDNA is an exciting development that will revolutionise the way we excavate and analyse our sites.

I cannot comment on the validity of the aDNA statistics and methods, however the conclusions are based on the findings and appear robust. The references are appropriate and the abstract, introduction and conclusions are clear and to the point.

My slight criticism to the text is that the first 2 paragraphs of the discussion repeat what has already been said previously in the text. I am not sure how necessary it is to do so.

Thank you for this suggestion. We have taken this into account and substantially shortened the discussion.

Finally, I find the presentation of the data, in particular the figures rather complicated, each been composed by several plates and, generally, difficult in the eye. I think the authors need to step back and decide their priorities on what exactly they want to present as their main findings and place the rest in the SI or remove them altogether.

Figure 1: There's too much information and many datapoints and demarkations to take in (black numbers, red numbers; red, blue circled numbers for different fossils; dashed lines; background shading; filled circles with red, blue, yellow, grey, empty circles; crosses, stars, diamonds, triangles). I suggest simplifying it. I don't particularly think providing modelled start and end ages on the left side of the columns is necessary, they clutter the figure even more. Maybe best include the total span of each archaeological phase (eMP, mMP, IUP). Plates a, b and c could remain in same figure, with (d) forming a new Fig. 3, possibly alongside plates (a) and (b) from Extended data Fig. 6?

We appreciate this feedback and recognize that some of the figures are complex. We are reluctant to break up the components in Figure 1, however, as this would require the reader to try and piece together the individual elements on their own. We feel strongly that it is important to provide the results for each sample in stratigraphic context (panels a–c), and panel d must be included because it defines the symbols used to denote the various hominin lineages in panels a–c. We have adjusted the caption of Figure 1 to make this clearer. It is also important to show the detailed age information in this figure, as several time gaps are present in the stratigraphic sequences in Main and East Chambers, and these gaps need to be taken into account when reading the record of sedimentation. For this reason, it would also be inaccurate to show only the total time span of each archaeological phase.

I'm afraid Extended data figures are way too complex.

Extended Data Fig. 1 is a repeat of what has already been published by Douka et al, and Jacobs et al. 2020. Why include it here when nothing in it is new? I suggest removing it or at best move to the SI. Equally the mammalian composition ext data figures are too many. A simplified version of these figures might be: Start with what is now Extended data Fig. 2 and turn it in Extended data Fig. 1, followed by Extended data Fig. 5 (which sort of synthesizes the findings of the paper). What appears as plate (c) in Extended data Fig. 6 should either be combined with, or follow, Extended data Fig. 5. Extended data Fig. 4, 7 and 10 appear to me SI material.

We acknowledge that some of the Extended Data figures are complex, but the numerical order of these items is dictated by their order of appearance in the text. Extended Data Figure 1 is based on the chronological data for Main and East Chambers presented in Jacobs et al. (2019), but it is the first time that these data have been placed side-by-side to create a common timescale. It is important to show this composite timescale, because it forms the basis for integrating the ancient DNA results for both chambers. Details of its construction are given in Methods (Common timescale for Main and East Chambers).

A few more specific comments:

- Lines 42-44: I suggest that the authors simplify this paragraph. Maybe best state the earliest ages for the occupation in each Chamber, and then state the span for the eMP, mMP and IUP irrespective to Chamber.

As the start dates and time span of each archaeological assemblage differ between the three chambers, we think it is important to be explicit about these differences in timing. We refer in the text to Extended Data Figure 1, which provides a visual display of the time spans of the various archaeological phases in Main and East Chambers.

- Line 44: Initial occupation starts in the East Chamber is in layer 15 which dates to ~200ka. Please check the original Jacobs et al. 2020 publication and correct accordingly.

We originally gave only the start date for mMP in East Chamber, and not the eMP. This may have been confusing, so we have now included the start date for the eMP in East Chamber (203 ± 14 ka) and made further light edits for clarification.

- Line 80: what's the median? If you include it in line 99 you should include it heretoo.

Thank you for catching this. We have now added this value on line 80.

- Line 165: I think it should be Last Interglacial (with capitals)

We have now edited this paragraph and replaced 'last interglacial' with Marine Isotope Stage 5, to avoid any ambiguity about the relevant time period.

Referee #3 (Remarks to the Author):

Zavala and colleagues present a fine-resolution reconstruction of mammalian and hominin occurrences and turnovers at Denisova Cave, from the Middle to Late Pleistocene, using sedimentary ancient DNA (sedaDNA) from >700 samples.

This is an excellent, enjoyable, and well-written paper. The careful experimental design, which includes systematic sampling and spatial replication within and between cave chambers, together with new methods for robustly identifying hominin mitochondrial lineages has allowed the authors to thoroughly explore the mammalian and hominin history of Denisova cave. The authors have additionally used the data set to examine DNA preservation against different sediment characteristics and time at this single locality. The distinct stratigraphic partitioning of the recovered hominin and mammalian DNA suggests that leaching and reworking - particular concerns for cave sediments - are not a major influence on the main results.

Of novelty, the authors identify a new neanderthal mitochondrial lineage and have developed a type of sedaDNA-based biostratigraphy. The findings of this latter development are supported by the skeletal record.

The methodology, data, analyses, and interpretations appear to be mostly sound. My concerns about the paper in its present form are as follows:

The entire manuscript, including the final paragraph, is focused on Denisova Cave. However, the approaches developed and showcased here have much wider applicability to other sites, taxa, and substrates. I suggest that the authors add a new concluding paragraph and sentence to the abstract to highlight the broad utility of their findings beyond this single locality.

Thank you for this suggestion. Due to length constraints, we were not able to extend the abstract, but we have added a concluding sentence to the end of the manuscript highlighting the broader implications of our work.

Consider inserting 'Pleistocene' into the title, as there is currently no temporal dimension mentioned.

This is another good idea. We have included the word 'Pleistocene' in the revised title, but it now exceeds the 75-character limit (by 9 characters), so the final decision rests with the editor.

L77-94: The authors test DNA preservation (as measured by amount, fragment size, and extent of 5' deamination) against time and multiple sediment characteristics. For two of these measures (amount and fragment size), the authors use taxonomically-combined data. For the deamination analyses, however, data are partitioned based on hominin and four mammalian families. What is the rationale for taxonomically splitting the deamination analyses and not, for example, the fragment size analyses? Given that this is sedaDNA derived from taxa with similar tissue and cellular structures, would it not be safe to assume that deamination and fragment size profiles from the sedaDNA are independent of taxonomic origin? In fact, the authors could test whether the four mammalian families have significantly different deamination profiles within samples. Assuming the results of this test are non-significant (as hinted at by e.g. Supplementary Figure 6), then I suggest that the deamination comparison analyses are performed on the taxonomically-combined data. This would increase statistical power and reduce the number of comparisons presented in Extended Data Figures 3 and 4, and SI Section 3.

The deamination rates do, indeed, vary slightly across mammalian families, although this may be hard to see

in the plots. Hyaenas consistently have the lowest deamination rates, ursids and canids have the highest, and bovids have intermediate values. A pairwise t-test for deamination rates among the four families shows that only ursids and canids do not have a significant difference in 5' deamination rates; all other p-values are $<2E-16$. For this reason, we decided to treat the mammalian families separately. We have added a short explanation to Supplementary Section 3, noting that these differences exist and that they may be due to actual differences in DNA degradation, mapping biases, or both.

L85-86,561,578: what does a 'significant increase' mean here? How was this tested or was a threshold used? The statistics used to determine the 'significant increase' referred to on lines 85–86 are provided in the caption to Extended Data Figure 4 (to which we refer in the main text). For the 'significant higher' statements on lines 561 and 578, we have added an explanation of the significance test (now lines 555 and 573).

L155-157: I thank the authors for formally correcting this error in their pilot study. One other mismatch in the literature exists; the layer from which Denisova-9 was recovered (Jacobs et al. 2019, Fig. 3: layer 12.2; Douka et al. 2019, Fig. 3: layer 12.3). The authors follow Jacobs et al. in Figure 1 - is this correct?

Thank you for drawing our attention to the layer mismatch of Denisova 9 between Jacobs et al. (2019) and Douka et al. (2019). This fossil was found in 2011, when layer 12 was not divided into sub-units, but its burial depth corresponds most closely to the upper part of layer 12.3 (as in Douka et al., 2019), rather than the lower part of layer 12.2 (as in Jacobs et al., 2019). We have added this information to the final paragraph of Supplementary Section 2.2 and refer to Douka et al. (2019) in the caption to Figure 1 where we mention the find locations of the hominin fossils. We have retained our correction of East Chamber 14/11.4 sediment sample misattribution in the text and in Supplementary Section 2.2.

L167-175: Modern human mitochondrial DNA was recovered from samples taken from the IUP and UP layers. For three of these samples, a single modern human contributor was determined (Supplementary Table 7). What are the mitochondrial haplogroups for these three samples? Assuming these assignments make phylogeographic and temporal sense, then this would provide additional supporting evidence for result authenticity.

We attempted to determine the haplogroup from these samples using HaploGrep2, but this resulted in low quality and conflicting haplogroup determinations, likely due to the low sequence coverage of the mtDNA genomes and/or the sequences falling basal to haplogroup-defining branches of the mtDNA tree. We have added a statement to this effect to the end of Supplementary Section 6.

L167-175: There seems to be only a single unreplicated sample with modern human mitochondrial DNA from the UP layer (Figure 1b). Do the authors consider this finding robust, given the potential for reworking from IUP layers? If not, I suggest the authors state this limitation.

There is no archaeological or stratigraphic reason to dismiss this result. It is generally believed that the UP was made by modern humans, so this result supports that notion. Single-grain OSL measurements suggest no mixing between the UP and IUP in this part of the deposit. It is more likely that the sediment type is less conducive to DNA preservation. Supplementary Section 3 shows that there is evidence for extensive phosphatization in this layer (Supplementary Table 2) and the pH of the two samples measured from layer 9 is also systematically lower than for samples from other layers (Supplementary Figure 4), similar to the situation for layer 9 in South Chamber. In both chambers, layer 9 is directly overlain by Holocene deposits, which we believe are the source of the post-depositional phosphates.

L173-175: Denisovan, neandertal, and modern human DNA are all found in layer 11.2 from the East Chamber, suggesting that all three groups were present in the area during the IUP. However, could this be an artefact of sediment reworking/mixing in this layer? Is there other evidence (chronological, stratigraphic, etc) to suggest that this is not the case?

The stratigraphy of this part of the deposit is complex and the dashed lines in the vicinity of these DNA samples in Figure 1a denote areas where the layer assignment is uncertain. It is tempting, therefore, to

speculate that the presence of all three hominin groups in layer 11.2 may be due to sediment mixing by biological or geological processes. We would need to carry out single-grain OSL analyses of each sediment sample from which DNA was recovered to exclude the possibility of small-scale mixing. Jacobs et al. (2019) measured one sample (DCE16-1) from layer 11.1 in the same profile and area as that sampled subsequently for DNA, and it showed no evidence for sediment mixing. Likewise, two samples measured from the 2014 profile (DCE14-10, layer 11.1 and DCE14-11, layer 11.2) also showed no evidence for significant mixing, but the latter sample yielded an age of ~60 ka, which is older than the age of ~45 ka typically associated with IUP deposits. In the absence of data directly associated with each of the samples from which mtDNA was extracted, we cannot make a definitive interpretation of the co-occurrence of Denisovan, Neanderthal and modern human DNA in layer 11.2 and their association with the IUP. Accordingly, we have included the following sentences (formerly in the 'Discussion') to indicate our lingering uncertainty over this result: "The situation in East Chamber is more complex: Denisovan, Neanderthal and ancient modern human mtDNAs were recovered from IUP layer 11.2, and Neanderthal and ancient modern human mtDNA from IUP layer 11.1 (Fig. 1a). Given these results and the recovery of two Denisovan fossils (*Denisova 3* and *4*) from layers associated with IUP assemblages, we cannot discount the possibility that, in addition to modern humans, Denisovans and Neanderthals may have been present during the period of IUP production⁹⁻¹¹."

L191-232: I think the authors can be bolder with their mammalian data, as they have effectively devised a sedaDNA-based biostratigraphy for Denisova Cave. For example, the author's could hypothesize that layer 22 from the South chamber is of early Middle Palaeolithic age based on the predominance of ursid (cave bear) DNA (Supplementary Figure 27; Supplementary Data File 1), which is characteristic of the eMP in the other two chambers (Extended Data Figures 8 and 9).

Thank you for this suggestion. We hope that our data will help make such inferences possible in future studies of the deposits in South Chamber. However, given the emphasis here on the results for Main and East Chambers, and the request of the editor to substantially shorten our manuscript, we have not elaborated on this point.

L208-220: Three mammalian families (ursids, elephantids, and hyaenids) were selected based on 'availability of complete mitochondrial genome sequences that cover the genetic diversity of extant and extinct species within a family' (pg45 of SI). However, this is also comparatively true for bovids and equids. Any reason to exclude these two families?

Another criterion was the grouping of sequences from the relevant species in monophyletic clades, which we have now added to the first paragraph of Supplementary Section 11. But we agree that there is certainly room for improving the analysis of faunal sequences on the species/population level, which is a line of research that we are currently pursuing.

From the three selected families, the authors exclude identified taxa that are unlikely to have been in the region (e.g. polar bear, Columbian mammoth). However, they include 'African and straight-tusked elephant' (Extended Data Figure 10; pg46 of SI). As African elephants are not known outside of Africa (including in the fossil record), the authors would be on safe ground to refer to this mtDNA group simply as 'straight-tusked elephant'.

Thank you for this suggestion. We have altered Extended Data Figure 10 and the relevant portion of Supplementary Section 11 to reflect this.

L224-225: for ecological context, state that this transition from an interglacial (MIS 7) to glacial (MIS 6) lead to climatic deterioration.

We have modified the text on lines 232–233 to read as follows: "...the climatic transition from an interglacial period (MIS 7) to a glacial period (MIS 6)." We feel that 'interglacial' and 'glacial' are sufficient to indicate the broad nature and direction of the climatic transition.

L227: the last interglacial is MIS 5e only (123-118 ka). Perhaps rephrase to 'during the climatic amelioration from MIS 6 to MIS 5' or similar.

To avoid ambiguity, we have removed 'last interglacial' from the text and rephrased lines 233–234 to read as follows: “A second turnover took place between about 130 and 100 (or 80) ka, during and after the climatic transition from MIS 6 to MIS 5.” The time spans of the various Marine Isotope Stages are shown in Fig. 2.

L241-242: How confident are the authors that their sedaDNA data are accurately reflecting relative abundance of large mammals? There seems to be a possible correlation with relative abundance of skeletal remains (Extended Data Figure 7), although the strength of this is not formally tested. Note that taphonomic and ecological factors, such as if the cave was periodically used as a predator den, could bias relative abundance estimates.

This is a very good point. We don't know the origin of the DNA recovered and it is likely that different mammals have deposited different quantities of DNA, due to differences in body mass, duration of occupation or behavior. This was hinted at in the first paragraph of “Ancient faunal mtDNA”, but we have now edited this section of the manuscript to point out more clearly that we do not expect the DNA to accurately reflect the relative abundance of large mammals present at the site—even though it is encouraging that some of the trends in the fossil record are also observed in the DNA record.

Figure 1: In panel a, should Denisova-9 be from layer 12.2 (following Jacobs et al. 2019) or 12.3 (following Douka et al. 2019)? In panel a, samples 251 and 244 should be modern human only (based on Extended Data Figure 6).

As noted above, we have assigned Denisova-9 to layer 12.3 (following Douka et al., 2019) and modified the side panel in Figure 1a accordingly. In panel a, there is a small signal of Neanderthal DNA in the libraries prepared from sample E251, in addition to modern human DNA. This signal is insignificant when the two libraries are analyzed separately (as shown in Extended Data Figure 6), but is significant when the data from both libraries are merged (as is the case in Figure 1). We noticed that two samples with modern human and Neanderthal DNA (E249 and E251) were not shown in their correct locations in this figure, so we have corrected them in the revised Figure 1a.

Figure 2: In panel a, % of biogenic silica is presented as an environmental proxy, but with no explanation in the figure legend. I assume the authors are using this as a proxy for local temperature? If so, is there good reason to use this record instead of the Greenland oxygen isotope record?

There seems to be conflicting warm or cold conditions inferred from the same time intervals between the Main and East chambers. Perhaps add a sentence describing the sources of uncertainty here.

The biogenic silica record from Lake Baikal is discussed in Jacobs et al. (2019) and the differences in some of the timings of cold and warm conditions inferred for Main and East Chambers are also discussed in that paper, so we have not repeated the information here. We treat this record as a proxy for regional annual temperature (following ref. 36) and have added a note to this effect in the figure caption. None of the Greenland ice cores have a continuous climate record that extend as far back as 300 ka: the oldest of them (the North Greenland Eemian Ice Drilling, NEEM, ice core) is just 128 ka at its base, so the Middle Pleistocene interval between 300 and 130 ka is not covered. As a long-term climatic record from southern Siberia, the Lake Baikal record is also more germane.

Extended Data Figure 1: perhaps add a panel with the position and assignment of hominin sedaDNA from the Slon et al. pilot study.

Thank you for this suggestion. We have now included trowel symbols next to the relevant layers in panels a and b, and noted in the caption that we have corrected the misattribution of the layer 14/11.4 sample in East Chamber. This makes the figure completely self-contained in terms of previous DNA analyses of sediment samples and hominin fossils from Main and East Chambers.

Extended Data Figure 2: what do the alphanumeric symbols at the left and top of panel a represent?

We have included the following explanation in the figure caption: “Grid coordinates for the excavation squares are shown along the top and left side of the plan, and the corresponding squares (each consisting of a Cyrillic letter and a number) are shown at the top of the stratigraphic profiles in Fig. 1a–c, Extended Data Figs. 3a,b, 5, 8 and 9, and Supplementary Figs. 1, 2 and 3b,d.”

Extended Data Figure 3: State here that Layer 11 is potentially IUP. The phosphate deformation deposits are

stratigraphically split into pdd-9 and pdd-12, but this division is not highlighted in panel a. Please also make the same correction to Supplementary Figure 3.

We have edited both figure captions to incorporate the IUP information, and the tentative assignments of individual samples to pdd-9 or pdd-12 are given on the first sheet of Supplementary Data File 1. We have not attempted to draw on the boundary between pdd-9 and -12 as these deposits have been extensively deformed and blurred by phosphatization.

Extended Data Figure 4: L379-381: are the test results for the average fragment size and number of fragments mixed up? The highest p-value is reported for average fragment size, but this shows a clearer decrease than number of fragments. Check also SI Section 3.1 (pg 9).

Thank you for catching these; we have corrected both instances.

Extended Data Figure 7: the panel letters are missing from the figure. In c and d, the x-axes are correct for aDNA, but should be 'percent of assigned remains' for skeletal. Some of the family names are truncated in c and d (note that this is also an issue for Supplementary Figures 21 and 22).

Thank you for drawing our attention to these. We have updated Extended Data Figure 7, as well as Supplementary Figures 21 and 22.

Extended Data Figure 10: I suggest renaming the hyaenids to be consistent with the other two families. For example 'Cave Hyaena, Hg B' instead of 'Haplogroup B'.

Good idea. We have changed the legend to 'Cave hyaena, haplogroup A' etc.

Minor:

L32-33: give the approximate age of the onset of the Initial Upper Palaeolithic

We now say "at least 45,000 years ago" to be consistent with the ages published for Denisova Cave, Tolbor-16 in Mongolia and Bacho Kiro Cave in Bulgaria, for example.

L38: give the age range in absolute time here as well, for readers not familiar with the ages of the Pleistocene and Holocene.

We think this might be confusing, as the optical ages of the cave sediments are listed immediately afterwards.

L60: archaic humans or archaic hominins?

Archaic hominins. We have now clarified this.

L68,81: check order of the Extended Data Figures

We have now fixed these, so that the Extended Data items are called out in numerical order.

L197: taxonomic family names should not be italicized

We have now fixed these.

L463: how far was the exposed profile cleaned back?

Immediately before collecting each sample, we cleaned back the exposed profile to a depth of approximately 1 cm using a sterilized scalpel blade. We have now added this information to line 454.

L514: how many copies of the control oligo were added to each sample?

This information is now included in current line 507.

L536-537: what read length was used? paired- or single-end?

This information has been added to current line 514.

Supplementary Information:

pg5, S2.1: layer 11.5 is also not exposed

Indeed – this has now been added (as well as layer 11.1 in Main Chamber, which was also not exposed in 2017).

pg7, S2.3: typo: 'ppd'

Thank you spotting this typo, now fixed.

pg9, S3.1: check order of statistical tests is correct (see above)

This has been corrected.

pg30, S7.2: Note that, in BEAST, groups can be selected to estimate tMRCA without enforcing monophyly.

This is a good point, so we have rephrased this sentence.

pg33, S8: typo: 'Supplementary Section X'

pg33, S8: typo: 'Using the full set of mtDNA genomes is used'

Both of these typos have been fixed.

pg33, S8: give reference for '[Vernot et al, in press]'

These details have been added to the reference list and we have also added a citation of the paper in the main text.

pg39, S9: consider re-writing the second paragraph to improve readability, perhaps just by giving the range of midpoint branch supports. The CI info could then be given in Supplementary Figure 24.

Thank you for these suggestions. We appreciate that this is complex section and hope to make the message as clear as possible. We have simplified the second paragraph, but decided to leave Supplementary Figure 24 as is, in order to not make it overly complex.

pg39, S9: typo: 'indicating the presence different'

pg46, S11: typo: '143'

pg46, S11: typo: 'beat'

We have fixed all of these typos.

Supplementary Figure 1: clarify in the legend that two separate samples were taken for two locations.

We have added a sentence to the figure caption to clarify this.

Supplementary Figure 3: more information is needed here on layers 11 and 22. Presumably 'deformed MP' means 'deformed Middle Palaeolithic'? And presumably layer 22 is older? How much younger is layer 11? And are the layers that include the 'phosphate deformation deposits' older or younger than layer 11? Please clearly indicate the Holocene deposits.

We have now defined 'MP' in the figure caption and added 'Holocene deposits' to panel a. The chronology of the South Chamber deposits is still a work in progress and the layer assignments are provisional at this stage. Our current understanding of the stratigraphy and chronology of these deposits is summarized in the penultimate paragraph of Supplementary Section 2, as follows: "To place the aDNA data for South Chamber in relative stratigraphic and chronological order, we have given tentative layer assignments to samples from the upper profile (pdd-9, layer 11 and pdd-12) and lower profile (dMP and layer 22). The latter samples are assumed to be stratigraphically lower—and therefore older—than those from the upper profile, but some dMP samples may overlap in time with those from pdd-12. The optical ages obtained for layers 22 and 11 indicate the approximate time span of sediment accumulation, with deposition of these layers estimated to have ended 269 ± 97 ka and started after 47 ± 8 ka, respectively¹."

Supplementary Figure 6: it is not necessary to also display the black outlier dots, given that measurement data are already overlain.

We have decided to keep the figures as they are, even if the black dots are not strictly necessary.

Supplementary Figure 16: the legend states: 'The chimpanzee branch used to root the tree is not shown.', but this is shown.

We have updated this figure, removing the chimpanzee branch to highlight the details of the hominin branches.

Supplementary Figure 20: please improve the resolution of this figure.

We have added a higher quality version of this figure.

Supplementary Figure 27: add a colour-to-mammalian-family lookup key to the figure.
This has been added.

Supplementary Data File 1, sheet 'General Sample Summary': please add the European Nucleotide Archive sample accessions here.

This information will be included in the final version of the paper. Supplementary Data File 2: check cell H103.

We have fixed this typo.

Supplementary Data File 3, sheet 'Mammalian mtDNA clade refs':
rename Arctotherium clade from 'Arctic bear' to 'shortface bear'. Arctotherium is the South American short-faced bear (Arctos is the Greek word for bear).

why are the elephantid clades in latin, whereas common names are used for the ursids and hyaenids?

In order to not confuse Arctotherium sp with Arctodus simus we have renamed Arctotherium as short-faced bear (South American) and Arctodus simus as short-faced bear (North American) in Supplementary Data File 3 and Supplementary Figures 32 and 43. The remaining corrections have now been incorporated.

Note that Supplementary Data Files 2 and 3 were uploaded in the wrong order. This has been corrected.

Reviewer Reports on the First Revision:

Referee #1 (Remarks to the Author):

The authors have fully addressed my comments in their revision.

Referee #2 (Remarks to the Author):

I am happy with the authors' response to previous comments. While they chose not to change the way they present their data to a more "reader-friendly" format, ultimately this is the authors' decision.

Referee #3 (Remarks to the Author):

The authors have thoroughly addressed all my previous concerns in the revised manuscript. I have no further comments.

Congratulations,

Pete Heintzman